# LaST$_0$: Latent Spatio-Temporal Chain-of-Thought for Robotic Vision-Language-Action Model

Zhuoyang Liu [* 1]  Jiaming Liu [* † 1]  Hao Chen [* 2]  Jiale Yu [1]  Ziyu Guo [2]  Chengkai Hou [1 3]  Chenyang Gu [1 4]
Xiangju Mi [1 4]  Renrui Zhang [2]  Kun Wu [3]  Zhengping Che [† 3]  Jian Tang [3]  Pheng-Ann Heng [2]
Shanghang Zhang [✉ 1]

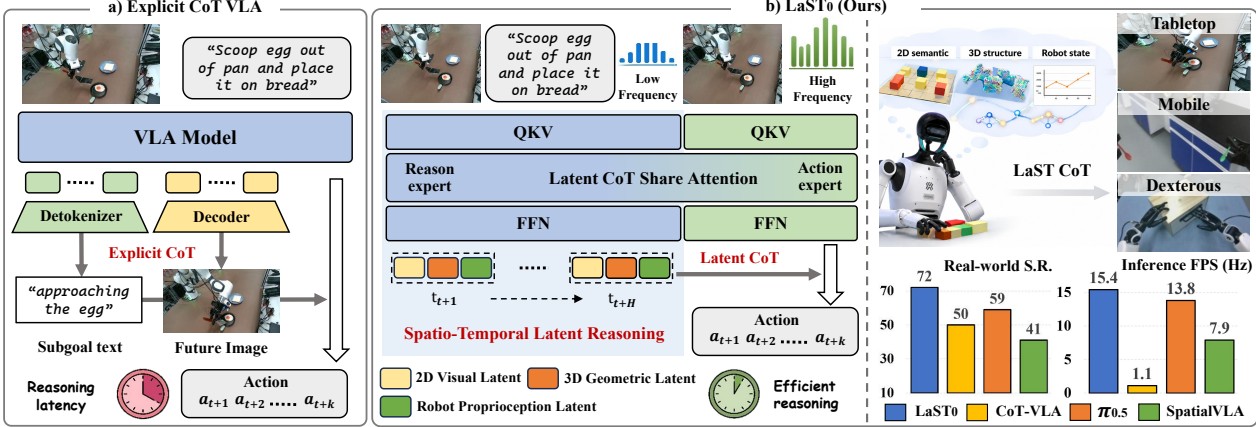

*Figure 1.* **Overview.** (a) Unlike previous VLA methods that explicitly generate linguistic reasoning traces or future visual observations, (b) we propose LaST$_0$, a framework that enables efficient reasoning before acting through a Latent Spatio-Temporal CoT. This latent CoT captures multimodal physical and robotic dynamics that are difficult to verbalize and propagates them over time to form temporally consistent reasoning. LaST$_0$ achieves SOTA performance across a wide range of tasks while enabling more efficient model inference.

## Abstract

Vision-Language-Action (VLA) models have recently shown strong generalization, with some approaches seeking to explicitly generate linguistic reasoning traces or predict future observations prior to execution. However, explicit reasoning typically incurs non-negligible inference latency, which constrains the temporal resolution required for robotic manipulation. Moreover, such reasoning is confined to the linguistic space, imposing a representational bottleneck that struggles to faithfully capture ineffable physical attributes. To mitigate these limitations, we

propose **LaST$_0$**, a framework that enables efficient *reasoning before acting* through a Latent Spatio-Temporal Chain-of-Thought (CoT), capturing fine-grained physical and robotic dynamics that are often difficult to verbalize. Specifically, we introduce a token-efficient latent CoT space that models future visual dynamics, 3D structural information, and robot proprioceptive states, and further extends these representations across time to enable temporally consistent implicit reasoning trajectories. Furthermore, LaST$_0$ adopts a dual-system architecture implemented via a Mixture-of-Transformers design, where a reasoning expert conducts low-frequency latent inference and an acting expert generates high-frequency actions conditioned on robotics-oriented latent representations. To facilitate coordination, LaST$_0$ is trained with heterogeneous operation frequencies, enabling adaptive switching during deployment. Across 10 real-world tasks spanning tabletop, mobile, and dexterous hand manipulation, LaST$_0$ improves mean success rates by 13%, 14% and 14% over prior state-of-the-art VLA methods, respectively. **Project page:** https://vla-last0.github.io/

*Equal contribution. †Project Lead. [1]State Key Laboratory of Multimedia Information Processing, School of Computer Science, Peking University, Beijing, China [2]The Chinese University of Hong Kong, Hong Kong, China [3]Beijing Innovation Center of Humanoid Robotics, Beijing, China [4]Simplexity Robotics-PKU Joint Laboratory, Beijing, China. Correspondence to: Shanghang Zhang <shanghang@pku.edu.cn>.

*Proceedings of the 43rd International Conference on Machine Learning*, Seoul, South Korea. PMLR 306, 2026. Copyright 2026 by the author(s).

# 1. Introduction

By inheriting the semantic understanding and common-sense reasoning capabilities of Vision-Language Models (VLMs) (Alayrac et al., 2022; Karamcheti et al., 2024; Deng et al., 2025a), Vision-Language-Action (VLA) models integrate rich pretrained knowledge with the low-level control capabilities of robotic policies (Brohan et al., 2023; Black et al., 2024; Intelligence et al., 2025). This integration endows robotic agents with a unified framework for interpreting human instructions and executing corresponding manipulation primitives in dynamic environments.

Rather than simply mapping observations to actions, recent advances in VLA models have been inspired by the Chain-of-Thought (CoT) reasoning paradigm in general VLMs (Guo et al., 2025). In this line of work, some approaches enhance manipulation stability and interpretability by explicitly generating linguistic reasoning traces or affordance representations (Ye et al., 2025; Li et al., 2025a; Zhao et al., 2025). In parallel, other studies seek to capture environmental dynamics by predicting future states (Tian et al., 2024; Zhang et al., 2025b; Wang et al., 2025c). Despite their demonstrated benefits, explicit CoT VLA methods remain constrained by two fundamental challenges in robotics manipulation. **On the one hand**, explicit reasoning typically incurs non-negligible inference latency. The autoregressive generation paradigm introduces inevitable computational overhead (Tan et al., 2025; Wang et al., 2025b), limiting the VLA model's ability to achieve real-time responsiveness. This latency further restricts VLA models from reasoning effectively along the temporal dimension, thereby undermining the temporal consistency required for closed-loop manipulation. **On the other hand**, explicit reasoning is often confined to the linguistic space, imposing a representational bottleneck that struggles to faithfully capture ineffable physical attributes. In contrast, robotic agents must reason about and interact with the physical world, which is essential for robust manipulation in a dynamic environment.

In this paper, we propose **LaST$_0$**, a dual-system VLA model that enables efficient reason-before-act behavior through a **La**tent **S**patio-**T**emporal Chain-of-Thought (CoT). As shown in Fig. 1, unlike prior explicit CoT-based VLA methods, LaST$_0$ performs reasoning in a compact latent space, enabling the capture of fine-grained physical and robotic dynamics that are difficult to verbalize, while supporting temporally coherent modeling. Specifically, we introduce a token-efficient latent CoT space that autoregressively predicts future latent tokens of 2D images, 3D point clouds, and proprioceptive states. As a result, the VLA model can implicitly model the semantic and geometric structure of physical dynamics, while forming an internal representation of the robot state, thereby capturing the relationship between the robot and its interactive environment. Meanwhile,

the latent CoT space is extended across future keyframes, enabling temporally consistent causal reasoning, which improves action coherence in closed-loop robotic manipulation. Although the proposed latent CoT is compact and encodes richer physical information, incorporating it into action generation still introduces additional inference overhead. Therefore, leveraging temporally extended latent conditions, we further propose a dual-system architecture implemented via a Mixture-of-Transformers (MoT) design. Specifically, two experts are integrated within a single VLA model: a slow reasoning expert, which performs low-frequency latent inference to capture spatio-temporal dependencies, and a fast acting expert, which generates actions conditioned on high-frequency observations and periodically updated latent representations. Through shared self-attention mechanisms, LaST$_0$ enables long-context interaction between the latent CoT space and the action space, thereby effectively coordinating deliberative reasoning with responsive control.

For the training procedure, both the latent reasoning expert and the action expert are initialized from the same pretrained VLM (i.e., Janus-Pro (Chen et al., 2025a)). We then perform large-scale pretraining of LaST$_0$ on diverse robotic manipulation datasets (Open X-Embodiment Collaboration et al., 2023; Khazatsky et al., 2024; Wu et al., 2025; Hou et al., 2025), ensuring seamless interaction between the two experts within a unified VLA model. During downstream training, we jointly optimize the two experts, with the action expert trained under heterogeneous fast-slow operating ratios, enabling the VLA model to adaptively select appropriate execution frequencies during deployment. For evaluation, we systematically assess LaST$_0$ on LIBERO (Liu et al., 2023) and 10 tasks in RLBench (James et al., 2020) simulation benchmarks and 10 complex real-world tasks spanning tabletop single- and dual-arm, mobile, and dexterous hand manipulation. LaST$_0$ reach 98.1% success rate on LIBERO benchmark, which also outperforms prior state-of-the-art VLA methods by 8% in RLBench and by 13%, 14%, and 14% across three real-world scenarios, respectively, while achieving a 14× speedup over previous explicit CoT VLA approaches. In addition, we validate the manipulation capability of LaST$_0$ on long-horizon real-world tasks, such as repeatedly scooping eggs from a pan while adapting to dynamic environmental changes. **Our contributions are summarized as follows:**

- We propose **LaST$_0$**, a unified VLA model that enables efficient reason-before-act behavior through a **La**tent **S**patio-**T**emporal CoT, performing reasoning in a compact latent space to capture fine-grained physical and robotic dynamics that are difficult to verbalize.
- We design a spatio-temporal latent CoT space, which autoregressively models future semantic, geometric, and proprioceptive information, allowing **LaST$_0$** to reason about physical dynamics in a temporally coherent manner.

- We introduce a dual system VLA architecture, implemented via MoT scheme, that coordinates low-frequency latent reasoning with high-frequency action generation, enabling real-time robotic manipulation.

## 2. Related Work

**Vision-Language-Action (VLA) Model.** VLA models are primarily driven by scaling robot demonstration data and adapting pretrained VLMs for robotic control (Belkhale et al., 2024; Li et al., 2023; Kim et al., 2024). To improve expressivity for continuous actions, recent VLA research has increasingly employed continuous generative policy heads. Diffusion-based VLA (Wen et al., 2024; Liu et al., 2025a; Fan et al., 2025) models complex action distributions through iterative denoising, while flow-matching formulations (Intelligence et al., 2025; Bjorck et al., 2025; Su et al., 2025) offer an alternative that can improve sampling efficiency and stability. Meanwhile, recent research equips VLA models with "reason-before-act" components to improve physical world reasoning. (Intelligence et al., 2025; Lin et al., 2025; Zawalski et al., 2025) adopts textual chain-of-thought (CoT) generation for future task planning and action generation. Subsequent work (Wang et al., 2025c; Zhao et al., 2025; Wen et al., 2025; Gao et al., 2025; Bu et al., 2025; Zhang et al., 2025b) extends generative text planning to future image prediction. Furthermore, (Liu et al., 2025b; Cen et al., 2025; Gu et al., 2025) introduces predictions of future multi-modal information. However, explicit reasoning typically incurs non-negligible inference latency. LaST$_0$ proposes a latent CoT strategy to efficiently capture fine-grained physical and robotic dynamics within a compact latent space.

**Latent CoT.** Recent work of VLM in the general domain (Chen et al., 2025b; Deng et al., 2024; Wang et al., 2025b; Li et al., 2025b; Yang et al., 2025) has explored latent CoT reasoning to address the limitations of explicit CoT on ineffable visual-spatial matching and high-cost generation. These methods perform multi-step inference directly in continuous latent spaces, allowing intermediate reasoning to be compact, implicit, and tightly integrated with downstream prediction. Beyond general-purpose VLMs, similar approaches have been adopted in the embodied intelligence domain, where textual CoT is particularly ill-suited for representing low-level signal generation. LCDrive (Tan et al., 2025) replaces language-based explanations with action-aligned latent rollouts for autonomous driving. Thinkact (Huang et al., 2025) compresses intermediate motion plans into compact representations. Our proposed LaST$_0$ is tailored for robotic manipulation, reasoning in a physically grounded latent space that jointly encodes semantic intent, geometric structure, and robot state, thereby capturing the embodied interaction between the robot and its environment.

## 3. Method

### 3.1. Preliminaries

**VLA Problem Formulation.** We formulate the robot manipulation task as a probabilistic sequence decision-making problem (Kim et al., 2024). At each timestep $t$, the policy receives a natural language instruction $l_t$ and visual observations $I_t \in \mathbb{R}^{H \times W \times 3}$ that capture the current environment. The objective of the VLA model $\pi_\theta$ is to generate an optimal action sequence $\mathbf{a}_{t:t+H}$ conditioned on the instruction $l_t$. We define the action space within the Special Euclidean group $SE(3)$. For single-arm configurations (e.g., Franka research 3), we employ a 7-DoF end-effector pose control mechanism, formulated as $\mathbf{a}_t \in \mathbb{R}^7$. Specifically, this control vector consists of 3-DoF for relative positional offsets ($[\Delta x, \Delta y, \Delta z] \in \mathbb{R}^3$), 3-DoF for rotation (represented as Euler angles [roll, pitch, yaw] $\in \mathbb{R}^3$), and 1-DoF for the gripper state (open/closed, $g \in \mathbb{R}^1$). For dual-arm configurations, we extend the action representation to 14 DoF by concatenating control signals to validate scalability; for mobile manipulation, we additionally estimate the base's linear and angular velocities. Additional robotic CoT reasoning preliminaries are provided in Appendix I.1.

### 3.2. LaST$_0$ Architecture

**Overview.** As illustrated in Fig. 2 a), we elaborate on the architectural of LaST$_0$. Our model is initialized on the Janus-Pro(Chen et al., 2025a), utilizing DeepSeek-LLM 1.5B as the backbone. To bridge the gap between reasoning and control, we transform this standard decoder-only transformer into a unified MoT dual-system architecture, constructing the "reason-before-act" paradigm. This design enables the model to effectively decouple the generation of slow, high-level latent reasoning from fast, low-level action execution, while maintaining seamless information flow through a shared attention mechanism.

**Vision Encoder.** For each input RGB observation $I_t \in \mathbb{R}^{H \times W \times 3}$ ($H = W = 384$), we employ SigLIP-Large(Zhai et al., 2023) to extract semantic features. The encoder yields a compact feature sequence $f_{\text{img}} \in \mathbb{R}^{B \times N_{\text{img}} \times d_v}$, where $B$ denotes the batch size, $N_{\text{img}}$ represents the sequence length, and $d_v$ is the embedding dimension. In our framework, these encoded features $f_{\text{img}}$ serve a dual purpose: the current frame acts as real-time contextual input to the MoT experts, while future frames provide ground-truth target embeddings for the visual component of the latent CoT.

**Point Cloud Encoder (training only).** To equip the model with fine-grained geometric perception and robust 3D spatial reasoning, we integrate a large-scale, pretrained point cloud encoder, Uni3D (Zhou et al., 2023b), which explicitly captures object geometry and spatial knowledge. Note that, unlike the vision encoder, the point cloud encoder is not

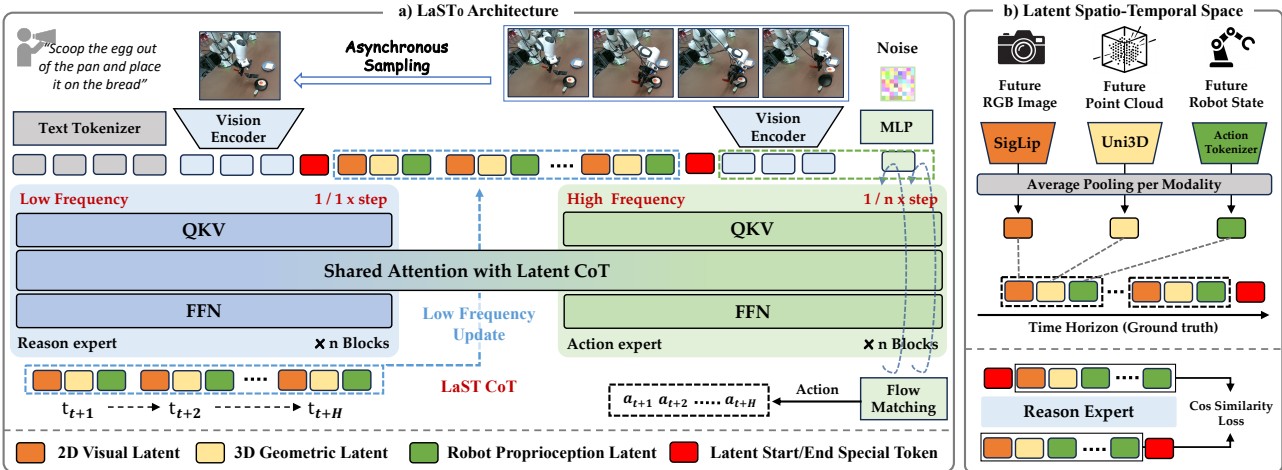

*Figure 2.* **Framework. a)** We propose LaST$_0$, a unified VLA model with a dual-system architecture. The model is implemented via a MoT scheme with two experts interacting through shared self-attention. The slow reasoning expert operates at a low frequency, taking images and text as input to construct the LaST CoT through autoregression. The fast acting expert operates at a higher frequency and generates actions via flow matching, conditioned on high-frequency observations and periodically updated latent representations. **b)** We design a spatio-temporal latent space, where pretrained modality-specific encoders extract features from future RGB images, point clouds, and robot states. These features serve as ground-truth latent CoT targets for supervising the reasoning expert.

used during inference. Instead, Uni3D is solely employed to encode ground-truth point clouds into compact 3D feature representations within the latent CoT space.

**Mixture-of-Transformers LLM Backbone.** We adopt DeepSeek-LLM 1.5B as our foundation, repurposing its 24-layer decoder-only transformer architecture into a unified dual-system policy. To efficiently support both high-level reasoning and low-level execution, we transform the standard backbone into a MoT architecture (Deng et al., 2025b). Unlike conventional transformers that apply a homogeneous set of weights to all tokens, our MoT design introduces task-specific parameter sets for all non-embedding components, including Feed-Forward Networks (FFN), attention projections $(W_Q, W_K, W_V, W_O)$, and Layer Normalizations, while maintaining a shared global self-attention context. This modification essentially yields two specialized experts residing within the same $d = 2048$ dimensional latent space, while a slow reasoning expert responsible for autoregressively synthesizing the Latent CoT embeddings $\mathcal{Z}$ from language and slow-stream visual inputs, and a fast acting expert dedicated to generating high-frequency actions $\mathbf{a}_t$. The design of different operating frequencies for the two experts is described in Section 3.4.

**MLP Components.** To further clarify the LaST$_0$ architecture, we describe the remaining auxiliary components, all of which are implemented as Multi-Layer Perceptrons (MLPs). First, a 3D Projector aligns point cloud features to the same dimension, serving as supervise targets for the Latent CoT. Regarding the action generation mechanism, given that our fast acting expert adopts a Flow Matching policy (Black et al., 2024), specific modules are incorporated to handle the continuous generative dynamics. For the action expert input,

we use a timestep MLP to encode the continuous time coordinate $t \in [0, 1]$, initialized with sinusoidal embeddings, and a noised-action MLP to project the perturbed action state. For the action expert output, a projector MLP is used to transform the predicted flow velocity field.

### 3.3. Latent Spatio-Temporal Chain-of-Thought

To capture fine-grained physical and robotic dynamics that are difficult to verbalize, while enabling efficient temporal modeling for manipulation, we construct a Latent Spatio-Temporal Chain-of-Thought (LaST CoT).

**Latent Embedding Construction.** To model temporal environmental dynamics, our latent representation encodes multimodal future states over a horizon $H$. As shown in Fig. 2 b), for each future timestep $k \in \{1, \dots, H\}$, we extract features from three complementary modalities to form a holistic physical representation. Future RGB frames $I_{t+k}$ are encoded into visual latents $z_k^v$ using the frozen SigLIP-Large encoder; simultaneously, future point clouds $P_{t+k}$ are processed by the Uni3D encoder to yield geometric latents $z_k^p$ capturing 3D spatial occupancy, while future robot states $\mathbf{s}_{t+k}$ are transformed into proprioceptive latents $z_k^s$ via an action tokenizer. Together, these representations enable the VLA model to implicitly model the semantic and geometric structure of physical dynamics, while maintaining an internal estimate of the robot's state, thereby capturing the interaction between the robot and its environment. To ensure high inference efficiency, we apply average pooling to compress the feature maps of each modality into a single representative token. This results in a compact set of embeddings $\{z_k^v, z_k^p, z_k^s\}$ for each step. We then organize these tokens in an interleaved, chronological order to preserve

causal physical dependencies:

$$\mathcal{Z}_{\text{GT}} = [\mathbf{z}_1^v, \mathbf{z}_1^p, \mathbf{z}_1^s, \mathbf{z}_2^v, \mathbf{z}_2^p, \mathbf{z}_2^s, \ldots, \mathbf{z}_H^v, \mathbf{z}_H^p, \mathbf{z}_H^s]. \quad (1)$$

The interleaved multimodal structure further encourages the model to learn the coupled dynamics across different modalities over time. Notably, the temporal granularity can be flexibly adjusted: we either adopt keyframe extraction as in (Shridhar et al., 2022) or use densely sampled frames, depending on task requirements. Moreover, by compressing high-dimensional sensory inputs into a latent sequence of length $3 \times H$, we avoid the prohibitive cost of decoding pixel-level images or long textual sequences. In Section 3.4, we introduce an asynchronous frequency design for the two experts, which further accelerates action generation.

**Sequence Structure.** To better organize LaST CoT reasoning and action generation, we introduce three special tokens: <latent_start>, <latent_end>, and a placeholder token <latent_pad>. The reasoning segment is structurally defined as a sequence bounded by the start and end tokens, with the intermediate positions reserved for the latent embeddings. During Training, we replace the intermediate <latent_pad> tokens with the ground-truth latent sequence $\mathcal{Z}_{\text{GT}}$. This allows the model to learn the transition dynamics via standard teacher forcing. While during inference, the model is initialized with <latent_start> followed by a sequence of <latent_pad> tokens. The slow reason expert then autoregressively generates latent embeddings, sequentially filling the positions of the <latent_pad> placeholders until the pre-defined horizon is filled. Horizontal length can be adaptively adjusted and its effect is further analyzed through ablation studies.

**Latent Supervision Strategy.** Although inference is performed in an autoregressive manner, we train the slow reasoning expert using continuous latent regression rather than discrete token likelihoods (Yang et al., 2025; Wang et al., 2025b). Specifically, the slow expert is trained to predict a sequence of latent reasoning states $\hat{\mathcal{Z}}$ in a next-step prediction manner, conditioned on preceding observations and context. Unlike conventional CoT supervision based on discrete token prediction, our latent targets consist of continuous, high-dimensional embeddings that encode future physical world states. To align the predicted latent with the ground-truth representations, we employ cosine similarity as the supervision objective. The loss is defined as:

$$\mathcal{L}_{\text{latent}} = \sum_{t=1} \left( 1 - \frac{\hat{\mathbf{z}}_t \cdot \mathbf{z}_t^{\text{GT}}}{\|\hat{\mathbf{z}}_t\| \, \|\mathbf{z}_t^{\text{GT}}\|} \right). \quad (2)$$

By maximizing directional alignment in the latent space, this objective encourages the model to anticipate future physical dynamics in a structured and compact manner.

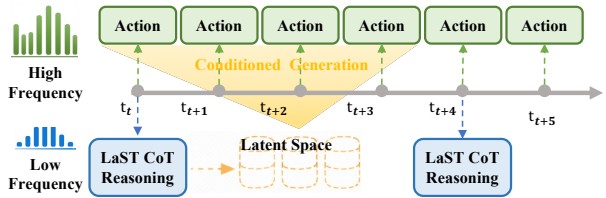

*Figure 3.* The reasoning expert performs low-frequency latent CoT reasoning to capture spatio-temporal dependencies, while the fast acting expert generates actions conditioned on high-frequency observations and periodically updated latent knowledge.

### 3.4. Dual-System Coordination

**Asynchronous Frequency Coordination.** To coordinate LaST CoT reasoning with high-frequency robotic control, we introduce an asynchronous frequency mechanism between the slow reasoning expert and the fast acting expert. As shown in Fig. 3, we decouple their operating frequencies using a set of update ratios $\kappa$ (e.g., $\kappa \in 2, 4, 8$). The slow expert is activated only at sparse keyframes ($t \bmod \kappa = 0$), where it performs autoregressive latent CoT reasoning. In contrast, the fast expert runs at the native control frequency and remains active at every timestep. Between consecutive keyframes ($t \bmod \kappa \neq 0$), the slow reasoning expert is held dormant, while the fast expert generates actions by conditioning on the most recent latent reasoning output. An interesting empirical finding is that increasing the temporal extent of the latent representations (e.g., by predicting multiple future keyframes) leads to improved performance in action prediction. For the inputs to the two experts, the slow reasoning expert receives the natural language instruction $l$ and the low-frequency observation $I_{\text{slow}}$, constructing the Latent CoT that encapsulates future physical dynamics. Conversely, the fast acting expert is optimized for rapid closed-loop feedback and receives only the high-frequency observation $I_{\text{fast}}$. Crucially, since our MoT architecture maintains a unified token sequence, the fast expert can efficiently attend to both the linguistic goal and the Latent CoT tokens via the shared attention mechanism. The description of the KV cache is provided in Appendix I.2.

### 3.5. Training Recipe

**Large-Scale Robotic Pretraining.** First, LaST$_0$ performs pretraining on a diverse corpus of over 400K trajectories aggregated from Open-X-Embodiment (Open X-Embodiment Collaboration et al., 2023), DROID (Khazatsky et al., 2024), RoboMIND (Wu et al., 2025), and other robotic datasets. Details are provided in Appendix B.

**Supervised Fine-Tuning (SFT).** We employ a joint SFT strategy that optimizes both the slow reasoning expert and the fast action expert. Specifically, the slow reasoning expert is trained by minimizing the Latent CoT regression loss $\mathcal{L}_{\text{latent}}$, aligning its latent representations with domain-

specific physical dynamics. In parallel, the fast acting expert is optimized using the standard Flow Matching loss $\mathcal{L}_{\text{flow}}$ for action denoising. Meanwhile, the action expert is trained with randomly mixed fast-slow operating ratios (e.g., 1:1, 1:2, 1:4), which exposes it to latent conditions updated at varying delays. As a result, LaST$_0$ can flexibly accommodate different reasoning-update frequencies at deployment and adaptively choose the fast-slow inference rate. In ablation studies, we find that training with mixed fast-slow operating ratios does not degrade performance; instead, it improves the model's robustness during inference.

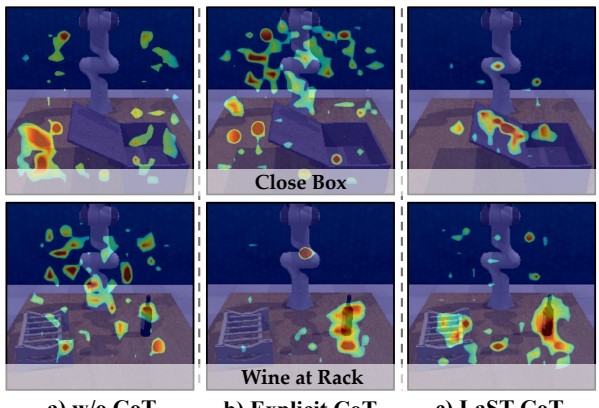

| a) w/o CoT | b) Explicit CoT | c) LaST CoT |

*Figure 4.* Attention heatmap visualizations from the last layer for three VLA models: (a) LaST$_0$ without CoT reasoning, (b) the explicit CoT in CoT-VLA, and (c) LaST$_0$ with LaST CoT.

## 4. Experiment

Section 4.1 evaluates the manipulation performance and inference efficiency of LaST$_0$ in simulation, while Section 4.2 conducts the ablation study of each component. Section 4.3 reports results on real-world tasks.

### 4.1. Simulation Experiment

**Benchmark Setup.** We evaluate LaST$_0$ on RLBench and LIBERO benchmarks to comprehensively validate the method across tasks in sundry scenarios. For the RLBench benchmark, we evaluate on a diverse set of 10 tasks, conducted in the CoppeliaSim simulation environment. All tasks are executed using a Franka Panda robotic arm with a single front-view observation. Demonstration data are collected by following pre-defined waypoints with motion planner (Sucan et al., 2012). Following the frame-sampling protocol adopted in (Shridhar et al., 2022), we construct a training dataset comprising 100 trajectories per task with keyframes. For the LIBERO (Liu et al., 2024) benchmark, our evaluation leverages its four specialized dataset suites: LIBERO-Spatial, LIBERO-Object, LIBERO-Goal, and LIBERO-Long. Each suite provides a standardized, highly structured corpus of 500 expert demonstrations, uniformly distributed across 10 distinct tasks.

**Training and evaluation protocol.** We train LaST$_0$ for 300 epochs during the Supervised Fine-Tuning (SFT) stage across both benchmarks, using the AdamW optimizer (Loshchilov & Hutter, 2017) on 8 NVIDIA A800 GPUs. LaST$_0$ is evaluated under a mixed fast-slow operating frequency with a reasoning-to-action ratio of 1:4. Further model and training hyperparameter details are provided in Appendix A. For RLBench benchmark, we benchmark LaST$_0$ against six representative state-of-the-art (SOTA) VLA models: OpenVLA (Kim et al., 2024), $\pi_{0.5}$ (Intelligence et al., 2025), CogACT (Li et al., 2024), SpatialVLA (Qu et al., 2025), CoT-VLA (Zhao et al., 2025), and HybridVLA (Liu et al., 2025a). For CoT-VLA, we reimplement the method on top of Janus-Pro (Chen et al., 2025a), the same foundation model used in our approach, and reproduce its explicit CoT reasoning to ensure a fair comparison. For LaST$_0$, the single front-view RGB image is resized to $384 \times 384$, accompanied by a point cloud uniformly subsampled to 1024 points, task instructions, and synchronized proprioception. Following (Goyal et al., 2023), we perform 20 rollout trials per task using the final checkpoint, repeat the evaluation across three random seeds, and report the mean success rate and variance. For the LIBERO benchmark, we further augment the set of baselines with OpenVLA-OFT (Kim et al., 2025), a SOTA method for this benchmark. Since the point cloud modality is unavailable in LIBERO, we remove it from the latent CoT content. Since demonstrations in LIBERO are densely sampled, we use a latent stride of 8 to increase the temporal coverage of the latent CoT. Each model uses the two RGB views (third-person and wrist), both resized to $384 \times 384$ as visual observations. In alignment with the official LIBERO protocol, each model is trained separately for each task suite, and we evaluate the final checkpoint on 500 trials per suite.

**RLBench benchmark results.** In Table 1, LaST$_0$-3.3B achieves a mean success rate of 82% across 10 RLBench manipulation tasks. In particular, LaST$_0$ surpasses the strongest existing methods, HybridVLA-7B (74%), $\pi_{0.5}$-3B (65%), and CogACT-7B (61%), by margins of 8%, 17%, and 21%, respectively. Beyond the overall average, LaST$_0$ attains the highest success rate on 7 out of 10 tasks, indicating consistent performance gains across diverse manipulation skills. These improvements primarily stem from the latent reasoning representations produced by the reasoning expert, which condition the action expert with compact latent states encoding future visual dynamics, 3D spatial structure, and robot proprioceptive information, enabling more stable and temporally coherent action generation. In terms of execution efficiency (without action chunking scheme), LaST$_0$ operates at an inference speed of 15.4 Hz, which is significantly faster than explicit CoT methods (CoT-VLA: 1.1 Hz) and remains competitive with $\pi_{0.5}$ (13.8 Hz). As shown in Fig. 4, we compare the attention heatmaps of LaST$_0$

*Table 1.* **Comparison of LaST$_0$ and baselines on RLBench benchmark.** All methods are trained in the multi-task setting (Shridhar et al., 2022), and we report mean success rates (S.R.). Inference speed is evaluated on an NVIDIA 4090 GPU.

| Models | Close box | Close laptop lid | Toilet seat down | Sweep to dustpan | Close fridge | Phone on base | Umbrella out | Frame off hanger | Wine at rack | Water plants | Mean S.R. ↑ & Var ↓ | Infer. speed ↑ |
|---|---|---|---|---|---|---|---|---|---|---|---|---|
| OpenVLA | 0.60 | 0.35 | 0.75 | 0.55 | 0.85 | 0.20 | 0.30 | 0.15 | 0.20 | 0.05 | 0.40 ±0.02 | 6.3 Hz |
| SpatialVLA | 0.80 | 0.70 | 0.85 | 0.20 | 0.80 | 0.15 | 0.25 | 0.40 | 0.15 | 0.30 | 0.46 ±0.03 | 7.9 Hz |
| CogACT | 0.90 | 0.80 | 0.95 | 0.50 | 0.85 | 0.50 | 0.55 | 0.45 | 0.30 | 0.25 | 0.61 ±0.04 | 9.8 Hz |
| CoT-VLA | 0.95 | 0.75 | **1.00** | 0.80 | 0.65 | 0.50 | 0.40 | 0.50 | 0.55 | 0.50 | 0.66 ±0.03 | 1.1 Hz |
| $\pi_{0.5}$ | 0.90 | **0.95** | 0.85 | 0.75 | **1.00** | 0.05 | 0.10 | **0.80** | 0.75 | 0.35 | 0.65 ±0.04 | 13.8 Hz |
| HybridVLA | 0.85 | **0.95** | **1.00** | **0.90** | **1.00** | 0.50 | 0.50 | 0.70 | 0.50 | 0.50 | 0.74 ±0.04 | 6.1 Hz |
| **LaST$_0$** | **0.95** | **0.95** | **1.00** | 0.80 | 0.85 | **0.75** | **0.75** | 0.70 | **0.85** | **0.60** | **0.82** ±0.03 | 15.4 Hz |

*Figure 5.* **Ablation study on key design choices of LaST$_0$.** We analyze (a) the importance of different latent modalities, (b) the number of tokens allocated per latent modality, (c) the temporal coverage in latent reasoning, and (d) the collaboration frequency between reasoning and action experts. Results are reported as average success rates across 10 RLBench tasks.

*Table 2.* **Comparison of LaST$_0$ and baselines on LIBERO benchmark.** The best results are highlighted in **bold**.

| Models | Spatial | Object | Goal | Long | Mean S.R. ↑ |
|---|---|---|---|---|---|
| OpenVLA | 84.7 | 88.4 | 79.2 | 53.7 | 76.5 |
| SpatialVLA | 88.2 | 89.9 | 84.6 | 55.5 | 78.1 |
| CogACT | 97.2 | 98.0 | 90.2 | 88.8 | 93.6 |
| CoT-VLA | 87.5 | 91.6 | 87.6 | 69.0 | 81.1 |
| $\pi_{0.5}$ | 98.8 | 98.2 | 98.0 | 92.4 | 96.9 |
| OpenVLA-OFT | 97.6 | 98.4 | 97.9 | 94.5 | 97.1 |
| **LaST$_0$** | **99.2** | **99.6** | **98.0** | **95.6** | **98.1** |

against variants without CoT and with explicit CoT (CoT-VLA). While other methods fail to aggregate features from the manipulated objects and the robot, LaST$_0$ exhibits a highly concentrated attention pattern, highlighting its superior spatio-temporal understanding.

**LIBERO benchmark results.** As shown in Table 2, LaST$_0$ consistently outperforms all baselines, achieving a SOTA mean success rate of 98.1%. In the Spatial and Object suites, which strictly evaluate generalization to novel layouts and objects, LaST$_0$ nearly saturates the performance, reaching 99.2% and 99.6% success rates, respectively. Furthermore, the LIBERO-Long suite presents a rigorous challenge for long-horizon execution. In this suite, LaST0 achieves a 95.6% success rate, outperforming strong baselines such as OpenVLA-OFT (94.5%) and $\pi_{0.5}$ (92.4%). This re-

sult demonstrates that our latent reasoning effectively enhances the model's ability to handle long-horizon tasks. Notably, the substantial improvement of LaST$_0$ over CoT-VLA (81.1%) highlights the advantage of compact latent reasoning in capturing physical dynamics and improving action execution stability.

### 4.2. Ablation Study

To validate the key design choices of LaST$_0$, we conduct comprehensive ablation experiments on 10 RLBench tasks.

**Importance of latent CoT modalities.** As shown in Fig. 5(a), we assess each latent modality by ablating individual components while keeping all other parameters fixed at their optimal settings (i.e., latent tokens = 1, temporal coverage = 4). When using only the image, point cloud, or robot state latent, the model achieves success rates of 74%, 76%, and 75%, respectively, indicating that each modality-specific latent provides a strong basis for action generation. The combination of multiple modality latents continues to provide additional performance improvements, even when the manipulation accuracy is already high. These results validate the importance of modeling comprehensive physical dynamics in the latent space, and further demonstrate that enabling the model to autonomously reason about the rela-

tionship between the robot and its interactive environment is effective for robotic manipulation.

**Number of tokens per latent modality.** As shown in Fig. 5 b), we study how the token budget allocated to each latent modality affects performance by varying the number of latent tokens while keeping other components fixed. When no latent token is used, performance drops substantially to 68%, indicating limited reasoning capacity. Introducing a single token for each modality leads to a sharp improvement (up to 82%), demonstrating that even a minimal latent representation is sufficient to establish an effective latent decision state. Further increasing the number of tokens yields no significant accuracy improvement, suggesting that high-level latent tokens can compactly encode physical information sufficient for effective reasoning. To further validate the impact of the number of latent tokens on representation quality, we conduct a PCA-based analysis of latent embeddings with varying token counts in Appendix F.1.

**Temporal coverage in latent reasoning.** As shown in Fig. 5 c), we investigate the effect of the temporal horizon used in latent reasoning by varying the number of future time steps encoded into the latent state. Performance improves consistently as the latent temporal coverage increases (from 68% to 82% when extending from 0 to 4 steps), indicating that incorporating longer temporal dependencies enables more informed latent decision states. Since adding further coverage beyond 4 steps does not significantly improve performance, we chose 4 steps as the final latent temporal coverage. The results suggest that extending the temporal horizon of latent prediction beyond that of action prediction is sufficient for improving action generation robustness. Due to our fast-slow system design, extending the temporal horizon of the latent space does not significantly affect action generation speed.

**Collaboration frequency between reasoning and action experts.** As shown in Fig. 5 d), varying the collaboration frequency has a clear impact on task success. A series of ratios such as 1:1, 1:2, and 1:4 achieve comparable performance (75-79%), while overly infrequent collaboration (1:8) leads to a relative drop to 74%. Finally, the mixed strategy trained by combining data from all collaboration ratios achieves the best performance (82%), using a 1:4 ratio during testing. These results indicate that our frequency joint training scheme enables more robust coordination between the reasoning and action experts across tasks.

**Impact of Architecture Components.** To further validate the significance of the MoT architecture, we conduct an additional ablation to evaluate our MoT dual-system design. Specifically, we employ a single backbone to implement both latent CoT and action generation, and simultaneously optimize the latent and action streams. This configuration reach a 74% success rate, while our method achieves an

8% higher. We also provide additional ablation studies on the impact of modality orders in LaST CoT and the impact of different latent temporal strides in dense action configurations in Appendix E, along with a PCA analysis of modality orders in Appendix F.3

### 4.3. Real-World Experiment

**Data collection.** We evaluated our method on a set of real-world manipulation tasks using both single-arm and dual-arm Franka robot setups, as well as AgileX mobile manipulation platforms (Robotics, 2024) and TienKung 2.0 humanoid dexterous hands (Center, 2024). Detailed configurations for different robots and tasks are provided in Appendix C and D. In the single-arm setting, we collect demonstrations for four tasks: 1) *wiping a whiteboard with an eraser*, 2) *pressing a stamp*, 3) *placing a dish on a rack*, and 4) *placing an egg on bread using a spatula*. In addition, we further evaluate a long-horizon setting on *placing egg* task, where the robot consecutively completes the full task three times while the positions of the manipulated objects continuously change during execution. For the dual-arm setting, two collaborative tasks were considered: 1) *scooping popcorn into a bowl* and 2) *opening a pot lid followed by picking corn from the pot*. For the mobile manipulation task, we evaluate the robot on two tasks: 1) *move to a table and stack plates* and 2) *sort spoons and place them into cabinet*, while for the humanoid robot, we evaluate on 1) *dexterous hand drawer opening* and 2) *dexterous hand place button*. Each task collects 200 teleoperation demonstrations. We show the task progress in Fig. 6.

**Training and evaluation details.** We train all policies following the same protocol as in simulation. We compare our method against three strong baselines: $\pi_{0.5}$ (Black et al., 2024), a state-of-the-art 2D VLA model; SpatialVLA (Qu et al., 2025), a state-of-the-art 3D VLA model; and CoT-VLA (Zhao et al., 2025), an explicit CoT-based VLA model that enhances action generation by predicting future visual observations. For fair comparison, all models use the same number of camera viewpoints, and each task is evaluated with 15 rollouts for 3 times under different positions.

**Quantitative and qualitative analysis.** As shown in Table 3, LaST$_0$ achieves the best overall performance on real-world manipulation tasks, with a mean success rate of 72% ($\pm 3$) on Franka platform (not including the long-horizon task), substantially outperforming SpatialVLA (41%, $\pm 2$), $\pi_{0.5}$ (59%, $\pm 4$), and CoT-VLA (50%, $\pm 2$). LaST$_0$ consistently delivers strong gains across a diverse set of tasks, particularly those requiring precise spatial reasoning and temporally coherent control. We further evaluate LaST$_0$ on a long-horizon manipulation task that requires one, two, and three consecutive successful executions within a single rollout. As shown in Table 3, LaST$_0$ maintains markedly higher

*Table 3.* **Comparison across real-world manipulation tasks.** We report success rates (S.R.) for standard single-arm and dual-arm tasks (Franka), mobile manipulation (AgileX), and dexterous manipulation (TienKung 2.0). Mean S.R. denotes the average success rate.

| Models | Wipe whiteboard | Press stamp | Place dish on rack | Place egg on bread | Scoop popcorn | Open pot pick corn | Mean S.R. ↑ | Place egg on bread | | | Arrange dishes | Sort spoon | Open drawer | Place button |
| --- | --- | --- | --- | --- | --- | --- | --- | --- | --- | --- | --- | --- | --- | --- |
| | | | | | | | | Step 1 | Step 2 | Step 3 | | | | |
| SpatialVLA | 0.60 | 0.67 | 0.30 | 0.20 | 0.27 | 0.40 | 0.41 | 0.20 | 0.07 | 0.00 | – | – | – | – |
| $\pi_{0.5}$ | 0.60 | 0.73 | 0.60 | 0.47 | 0.53 | **0.60** | 0.59 | 0.47 | 0.20 | 0.07 | 0.47 | 0.20 | 0.67 | 0.53 |
| CoT-VLA | 0.53 | 0.60 | 0.66 | 0.33 | 0.33 | 0.53 | 0.50 | 0.33 | 0.13 | 0.07 | 0.33 | 0.13 | 0.53 | 0.40 |
| LaST$_0$ | **0.73** | **0.93** | **0.80** | **0.66** | **0.66** | 0.53 | **0.72** | **0.66** | **0.47** | **0.33** | **0.67** | **0.27** | **0.87** | **0.60** |
| Embodiment | Franka Emika Panda | | | | | | | | | | AgileX Mobile | | Tienkung 2.0 | |

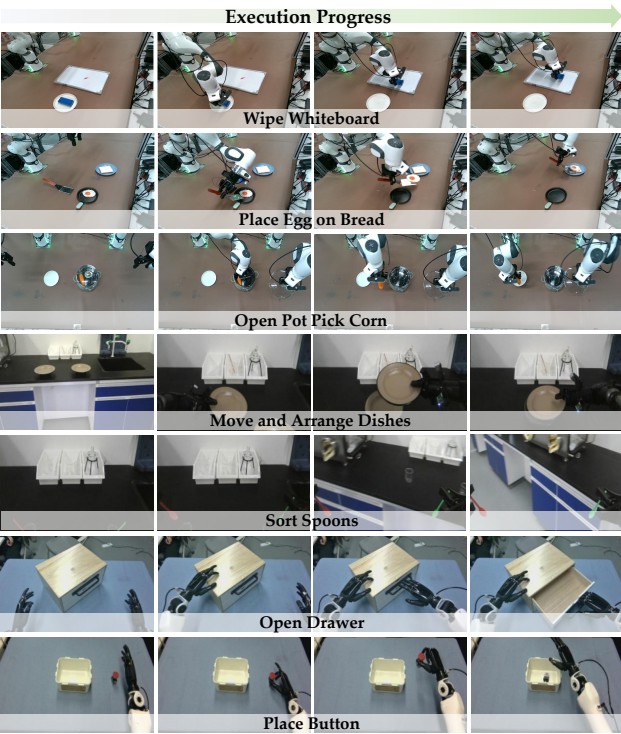

*Figure 6.* Visualization of Real-World Task Execution.

success rates across all stages (0.66 → 0.47 → 0.33) compared to $\pi_{0.5}$ (0.47 → 0.20 → 0.07), with the performance gap widening as the horizon increases. This trend indicates that LaST$_0$ is better able to preserve coherent latent representations of task progress and environment state over extended horizons. Beyond tabletop manipulation, LaST$_0$ consistently performs precise navigation and coordinated dual-arm control in mobile manipulation tasks, demonstrating that its latent spatio-temporal reasoning generalizes to larger action spaces beyond tabletop settings. For humanoid robots with higher degrees of freedom and dexterous hands, LaST$_0$ successfully handles complex articulated object manipulation, indicating that its reasoning and action generation capabilities are not constrained by embodiment complexity. We omit comparison with SpatialVLA due to the poor depth estimation. We show more comprehensive visualizations in Appendix G and supplementary video, and failure cases in Appendix H.

## 5. Conclusion

We introduced LaST$_0$, a dual-system VLA model that enables efficient reason-before-act behavior for robotic manipulation through a Latent Spatio-Temporal Chain-of-Thought (LaST CoT). By shifting reasoning from explicit traces to a compact latent space, LaST$_0$ overcomes the latency and representational bottlenecks inherent in prior CoT VLA approaches, while preserving the ability to model fine-grained physical dynamics essential for closed-loop control. Central to our framework is a token-efficient spatio-temporal latent representation that autoregressively captures future semantic, geometric, and proprioceptive dynamics. Building upon this LaST CoT, we further proposed a fast-slow dual-system implemented via a MoT, which decouples low-frequency deliberative reasoning from high-frequency action generation. We believe LaST$_0$ represents a step toward more physically grounded reasoning in robotic foundation models.

## 6. Limitations and Future Work

While LaST$_0$ enables efficient closed-loop control via latent reasoning, several avenues remain for future exploration. First, due to the scarcity of public robotic datasets, our pretraining coverage of complex mobile and dexterous manipulation is limited. Second, handling complex object interactions, remains challenging. We aim to explicitly enforce physical constraints and 3D relational graphs within the Latent CoT. Finally, we will explore reinforcement learning for post-training to enhance the robustness. Jointly optimizing latent reasoning and action generation will be crucial for scaling LaST$_0$ to longer-horizon, complex tasks.

## Acknowledgement

This work was also supported by the National Natural Science Foundation of China (62476011). This work was supported by the National Natural Science Foundation of China (625B2007). This work was supported in part by the InnoHK initiative of the Innovation and Technology Commission of the Hong Kong Special Administrative Region Government via the Hong Kong Centre for Logistics Robotics. This work was also supported by Beijing Innovation Center of Humanoid Robotics.

## Impact Statement

This paper presents work whose goal is to advance the field of Machine Learning. There are many potential societal consequences of our work, none which we feel must be specifically highlighted here.

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

## A. Model and Training Details

Detailed model architectures and training hyperparameters for LaST$_0$ are summarized in Table 4.

*Table 4.* Model and Training Configurations for LaST$_0$.

| Parameter | Value |
|---|---|
| *Reasoning Expert (LaST CoT)* | |
| Base Foundation Model | Janus-Pro 1.5B |
| Hidden Feature Dimension | 2048 |
| Transformer Layers | 24 |
| Max Sequence Length | 2048 |
| Latent Tokens per Step | 1 |
| Latent Stride (for Dense Data) | 8 |
| Spatio-Temporal Horizon | 4 |
| Attention Implementation | SDPA |
| *Action Expert (Flow Matching)* | |
| State Dimension | 7 & 14 |
| Action Dimension | 7 & 14 |
| Action Chunk | 1 & 16 |
| Training Timestep Sampling | $Beta(1.5, 1.0)$ |
| Num Inference Timesteps | 10 |
| *Training Configurations (SFT)* | |
| Optimizer | AdamW |
| Peak Learning Rate | $1 \times 10^{-4}$ |
| Min Learning Rate | 0 |
| LR Scheduler | Cosine with min LR |
| Weight Decay | 0 |
| Warmup Ratio | 0 |
| Gradient Clipping | 1.0 |
| GPU Type | NVIDIA A800 |
| Number of GPUs | 8 |
| Deepspeed Zero Stage | 1 |
| Per Device Batch Size | 8 |
| Gradient Accumulation Steps | 1 |
| Mixed Precision Training | bf16 |

## B. Large Scale Pre-training Datasets

To ensure LaST$_0$ inherits a robust foundation of motor primitives and physical common sense, we curated a diverse corpus of 400K trajectories (28M frames) from the Open-X-Embodiment (Open X-Embodiment Collaboration et al., 2023), DROID(Khazatsky et al., 2024), and RoboMIND(Wu et al., 2025) repositories. Table 5 lists the detailed proportions of each dataset used. Notably, beyond following prior VLA works in applying data quality filtering (Kim et al., 2024; Liu et al., 2025b), we additionally ensure that all robot state annotations are accurate and physically consistent. Specifically, to empower the slow reasoning expert with early geometric awareness, we utilized VGGT (Wang et al., 2025a) to generate synthetic 3D point clouds for all pretraining frames. These generated point clouds serve as the initial 3D geometric latent

($z^p$) inputs within the LaST CoT, allowing the model to learn spatial occupancy and environment dynamics even in the absence of real-world depth sensors during pretraining. This strategic alignment ensures a seamless transition to the full multimodal LaST CoT space during fine-tuning. This pretraining stage optimizes the model on broad robotic datasets to establish a shared representation space, enabling seamless interaction between reasoning and execution within the unified VLA framework.

*Table 5.* **Datasets used for pre-training.** The names of selected datasets for large-scale pretraining and their sampling ratios (%).

| Dataset | Ratio (%) |
| --- | --- |
| BC-Z (Jang et al., 2022) | 7.54 |
| Berkeley Autolab Ur5 (Chen et al.) | 0.35 |
| BridgeV2 (Ebert et al., 2022; Walke et al., 2023) | 20.93 |
| CMU Stretch (Mendonca et al., 2023) | 0.02 |
| DLR Sara Grid Clamp (Padalkar et al., 2023) | 0.02 |
| DROID (Khazatsky et al., 2024) | 4.82 |
| Dobb-E (Shafiullah et al., 2023) | 0.18 |
| FMB Dataset (Luo et al., 2023) | 1.50 |
| Fractal (Brohan et al., 2022) | 13.67 |
| Furniture Bench (Heo et al., 2023) | 0.09 |
| Jaco Play (Dass et al., 2023) | 0.19 |
| Kuka (Kalashnikov et al., 2018) | 20.22 |
| Language Table (Lynch et al., 2023) | 7.72 |
| Maniskill (Gu et al., 2023) | 5.26 |
| Nyu Franka Play (Cui et al., 2022) | 0.24 |
| Robo-Net (Dasari et al., 2020) | 11.53 |
| Roboset (Kumar et al., 2023) | 3.21 |
| RoboTurk (Mandlekar et al., 2018) | 0.70 |
| Stanford Hydra (Belkhale et al., 2023) | 0.20 |
| Taco Play (Rosete-Beas et al., 2022; Mees et al., 2023) | 1.26 |
| Toto (Zhou et al., 2023a) | 0.17 |
| Utokyo Pr2 Fridge (Oh et al., 2023) | 0.01 |
| Utokyo Pr2 Tabletop (Oh et al., 2023) | 0.04 |
| Utokyo Xarm Pap (Matsushima et al., 2023) | 0.04 |
| RoboMIND (Wu et al., 2025) | 0.2 |

## C. Real-world Set-up

### C.1. Franka Robot Setups

The physical deployment of LaST$_0$ utilizes a modular robotic infrastructure designed to provide the rich multi-modal feedback necessary for deliberative reasoning and responsive control. As shown in Fig. 7a, for single-arm configurations, we deploy a Franka Research 3 (FR3) manipulator paired with a ROBOTIQ adaptive gripper. The perceptual backbone consists of two Intel RealSense D455 cameras: a stationary third-person unit providing the right-front perspective required for the Slow Reasoning Expert's spatio-temporal planning, and a wrist-mounted unit for high-frequency visual servoing.

As shown in Fig. 7b, in dual-arm scenarios, the architecture scales to two parallel FR3 arms with identical end-effector and haptic configurations. This bimanual setup expands the perceptual suite to a three-camera array, including an additional

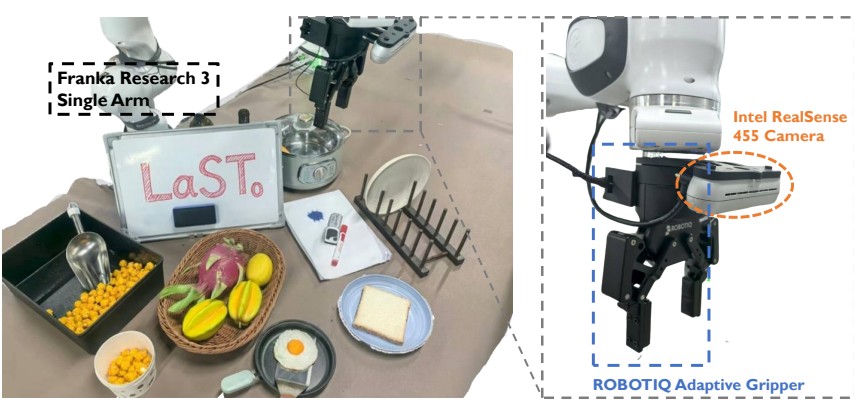
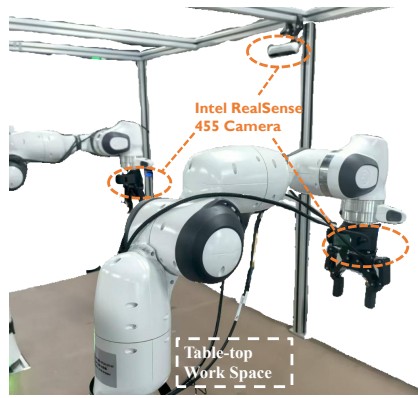

*(a)* Single-arm setup of Franka robot.        *(b)* Dual-arm setup of Franka robot.

*Figure 7.* Robot experiment setups of Franka Panda Arms.

front view camera and dual wrist cameras to capture the complex, synchronized spatio-temporal dependencies essential for collaborative manipulation. All demonstrations were collected using the Gello platform (Wu et al., 2024), with 200 demonstrations per task. The robot arm and gripper control methods are described in the preliminaries of the main paper.

### C.2. AgileX Mobile Manipulation Setups

As shown in Fig. 8a, for Agilex Mobile Manipulation tasks, we employ the Agilex Cobot Magic platform, including four Agilex Piper arms and a Tracer mobile base. Each of the puppet arms are equipped with an Orbbec Dabai camera at its wrist to capture RGB images. In addition, an Intel RealSense 515 camera is used on the robot's head view to obtain head observations. Three views of RGB images are used in total during training. For this platform, we adopt a unified 20-DoF action space $\mathbf{a}_m \in \mathbb{R}^{20}$. The first 14 dimensions represent the 6-DoF joint position (delta angles) and 1-DoF gripper state for the right (R) and left (L) arms, respectively. The final 6 dimensions control the mobile chassis, consisting of 3-DoF linear velocities and 3-DoF angular velocities:

$$\mathbf{a}_m = [\Delta\theta_{1:6}^R, g^R, \Delta\theta_{1:6}^L, g^L, \mathbf{v}_{lin}, \boldsymbol{\omega}_{ang}] \in \mathbb{R}^{20} \tag{3}$$

### C.3. TienKung Humanoid Dexterous Hand Setups

As shown in Fig. 8b, for the TienKung Dexhand Manipulation task, we employ the TienKung 2.0 platform, which consists a TienKung 2.0 Humanoid Robot, two ROHand AP001 R01 Dexterous hands and a Orbbec Gemini 336 camera on its head. Only one RGB image is used as image observation during training. We adopt a unified 26-DoF action space $\mathbf{a}_d \in \mathbb{R}^{26}$. This vector concatenates control signals for the right and left arms sequentially, with each arm accounting for 13 dimensions. Specifically, within each 13-DoF block, the first 7 dimensions represent the delta joint angles of the arm, followed by the 6 dimensions for the delta joint angles of the dexterous hand:

$$\mathbf{a}_d = [\Delta\theta_{1:7}^R, \Delta\phi_{1:6}^R, \Delta\theta_{1:7}^L, \Delta\phi_{1:6}^L] \in \mathbb{R}^{26}$$

Notably, to accommodate changes in action dimensions, only the noised-action MLP and the final MLP requires retraining to project noise to LLM's embedding space and project hidden state to final action, respectively.

## D. Self Collected Data

Building upon our real-world experiment setup, we evaluate LaST$_0$ across 10 representative tasks. These scenarios are specifically designed to validate the model's ability to balance high-level deliberative reasoning via the LaST CoT with responsive, high-frequency execution.

### D.1. Franka Emika Panda Arms

*1. Wipe whiteboard.* This task requires the robot to pick up an eraser on the table and clear colored blocks through visual recognition. Success depends on the model's ability to maintain a stable spatial trajectory and ensure the erasing path

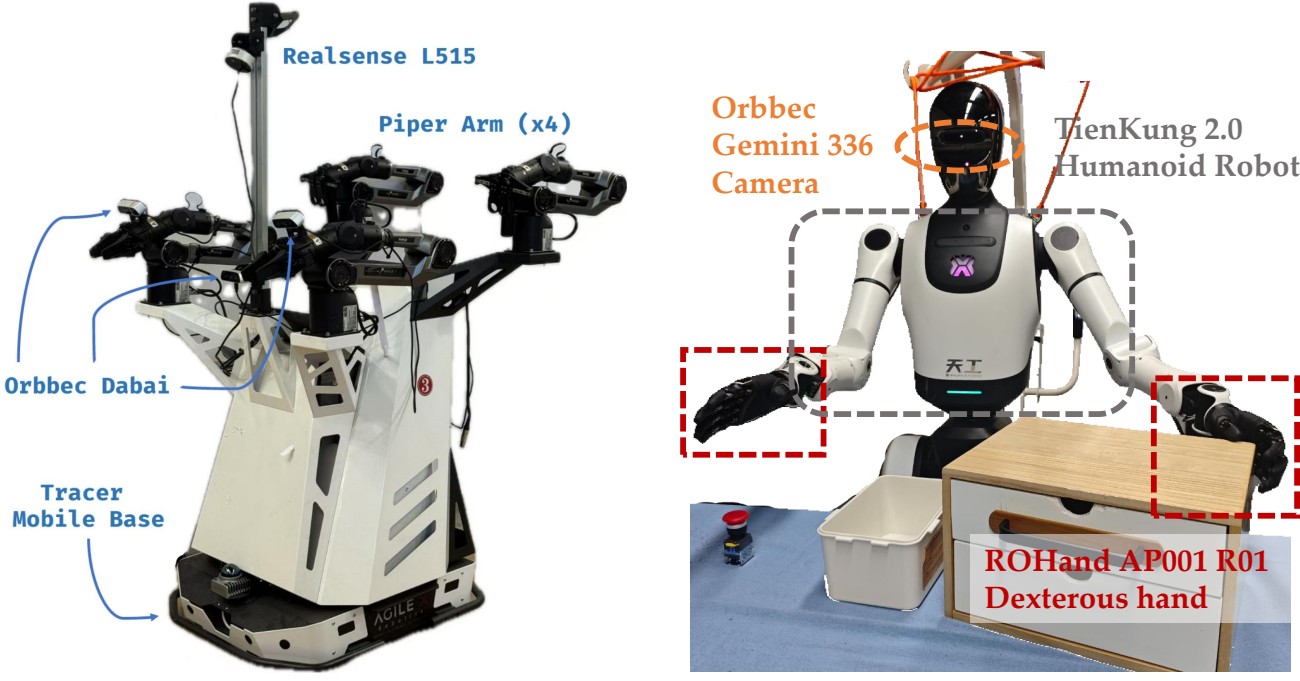

*(a)* Agilex Robot hardware configuration.

*(b)* Tienkung 2.0 hardware configuration.

*Figure 8.* Robot experiment setups of Agilex Mobile Manipulation Tasks and Tienkung Dexterous hand Manipulation Tasks.

accurately covers the target blocks.

*2. Press stamp.* The robot is required to establish a stable grasp on a stamp and execute a vertical press onto paper. This task evaluates the model's ability to maintain a consistent execution plan even when the critical contact point is visually occluded during the final press.

*3. Place dish on rack.* This task requires the robot to grasp a plate and perform a large 6-DoF rotation to insert it into a narrow rack. This demands highly accurate spatial perception and the ability to predict complex rotational trajectories for upright placement.

*4. Place egg on bread.* This task requires the robot to pick up the spatula, lift the egg, and place the egg onto the bread. The model must reason about the precise relative positioning between the spatula tip and the pan to slide under and lift the egg without failure.

*5. Scoop popcorn into bowl.* This collaborative task involves one arm scooping with a bucket while the other arm holds a bowl, then placing the bowl with popcorns on the plate. It validates the model's capacity for precise temporal synchronization and spatial coordination between two independent action streams.

*6. Open pot pick corn.* A long-horizon sequence involving lid removal, object retrieval, and lid replacement. This task evaluates the model's planning depth and collision-avoidance capabilities within a restricted workspace requiring seamless bimanual interaction.

### D.2. Agilex Mobile Manipulation Tasks

*1. Arrange dishes.* This task requires the robot to move front to the table and arrange two dishes collaboratively with two arms. Completing the task requires coordination between the chassis position and the robotic arm position, demonstrating the model's multi-dimensional control capabilities for long-horizon planning.

*2. Sort spoons.* The robot is required to pick up two spoons with two arms respectively and move to the cabinet, then place the spoons in it. The process of picking up the spoons requires the model to have precise manipulation capabilities, while the subsequent movement requires the model to understand spatial relationships in an open environment.

### D.3. Tienkung 2.0 Humanoid Dexterous Hand Task

*1. Open drawer.* This task requires the robot to handle the drawer with the left hand and open drawer with the right hand. The right hand needs to be precisely positioned on the drawer handle to successfully pull out the drawer, requiring a high degree of fine-grained manipulation of the model.

*2. Place button.* This task requires the robot to pick up the button with dexterous hand and place it into the box. The model needs to accurately handle the relationship between each finger and the complex surface structure of the object.

## E. Additional Ablation Study

### E.1. Impact of Modality Orders in LaST CoT.

We conduct an ablation study on modality ordering on the 10-task RLBench benchmark. In the original design, the latent sequence follows an environment-to-robot cognitive flow: [Visual, Geometric, State]. To rigorously evaluate whether this ordering affects the quality of latent representations and manipulation performance, we examine multiple permutations of these modalities. As shown in table 6, LaST$_0$ remains robust across different permutations, while the original ordering consistently achieves the best manipulation performance. This validates the design choice of the [Visual, Geometric, State] sequence, which aligns with a natural embodied reasoning process: first establishing global semantic context, then refining it with spatial structure, and finally grounding it in the robot's physical state before action prediction.

*Table 6.* Success Rate across different latent modality orderings.

| Modality Ordering | Success Rate (%) |
| --- | --- |
| [Visual, Geometric, State] (LaST$_0$) | 82 |
| [State, Geometric, Visual] | 78 |
| [Geometric, Visual, State] | 79 |
| [Visual, State, Geometric] | 81 |
| [State, Visual, Geometric] | 80 |
| [Geometric, State, Visual] | 79 |

### E.2. Impact of Different Latent Temporal Strides.

We conduct an additional ablation study to investigate the impact of using keyframe-based latent representations versus densely sampled latent representations in LaST CoT for manipulation. We perform experiments on the LIBERO benchmark that operates on high-frequency, dense observations across diverse tasks. For these experiments, we introduce a new hyperparameter **latent stride**, which enforces temporal sparsity in the latent reasoning process while maintaining dense action outputs. We validate the effectiveness of this design through an ablation study on 10 tasks from the LIBERO-Spatial benchmark, analyzing the impact of latent temporal spacing on performance. As shown in the table 7, maintaining a sparse latent construction (latent stride = 8) is critical in continuous prediction settings, as it enables the reasoning expert to capture more informative long-horizon dynamics. These results show that sparse latent construction (keyframe-like setting) is not only sufficient but also beneficial, as it enables the model to focus on long-horizon dynamics rather than redundant frame-level details.

*Table 7.* Impact on different latent strides in dense action settings.

| Latent Stride | 1 | 2 | 4 | 8 | 16 |
| --- | --- | --- | --- | --- | --- |
| Success Rate (%) | 97.4 | 98.0 | 98.8 | 99.2 | 99.0 |

## F. Additional Latent Space Analysis

In this section, we provide further quantitative and qualitative analysis of the Latent Spatio-Temporal Chain-of-Thought (LaST) space using Principal Component Analysis (PCA). Our goal is to analyze the representation dynamics within the latent reasoning space and its direct impact on action generation.

## F.1. Representation Dynamics in Latent Reasoning Space.

We conduct PCA on the latent CoT tokens extracted from the RLBench 10 tasks data. By analyzing the consecutive frame inter-distance in the projected PCA space, we measure how rapidly the latent representations evolve over time. As shown in Table 8, the latent space yields well-separated clusters across all modalities (Image, Point Cloud, and State), reflecting clear inter-token discrimination that captures temporal dynamics. We further relate this analysis to our token-number ablation, where we vary the number of tokens allocated per modality at each time step. The results indicate that increasing the number of tokens beyond 1 yields only marginal changes in inter-token distance. This quantitative observation strongly aligns with the experimental findings in Figure 5, corroborating that a single token per modality is sufficiently expressive to capture essential physical dynamics and temporal variations.

*Table 8.* **Consecutive Frames Inter-Distance by PCA.** Distances are measured across different time steps for Image, Point Cloud, and State modalities with varying token numbers.

| Time Steps | Image | | | Point Cloud | | | State |
|---|---|---|---|---|---|---|---|
| | 1 Token | 4 Tokens | 16 Tokens | 1 Token | 4 Tokens | 16 Tokens | 1 Token |
| Step ($t$ vs $t+1$) | 0.68 | 0.70 | 0.63 | 0.20 | 0.20 | 0.29 | 0.57 |
| Step ($t$ vs $t+2$) | 0.78 | 0.79 | 0.69 | 0.41 | 0.33 | 0.42 | 0.19 |
| Step ($t$ vs $t+3$) | 0.97 | 0.90 | 0.88 | 0.52 | 0.42 | 0.55 | 0.50 |
| Step ($t$ vs $t+4$) | 1.38 | 1.30 | 1.22 | 1.09 | 1.15 | 1.16 | 0.83 |

## F.2. Impact of Latent CoT on Action Generation.

To assess the effectiveness of the proposed latent reasoning framework, we perform a PCA-based quantitative analysis directly on the features of the generated action tokens. By projecting these features into a 2D space, we measure the inter-class distances between different motion categories. As demonstrated in Table 9, LaST$_0$ produces a significantly larger inter-class separation (1.34) compared to the baseline without Latent CoT (1.03). This indicates that our latent reasoning representations are more discriminative and structured, directly facilitating more stable and consistent downstream action generation.

*Table 9.* **Action Inter-class Distance.** Comparison of action feature separability with and without the proposed LaST$_0$ framework.

| Model Architecture | Action Inter-class Distance |
|---|---|
| w/o LaST CoT (Baseline) | 1.03 |
| **LaST$_0$** | **1.34** |

## F.3. Modality Contribution to Action Separability.

To explicitly analyze the contribution of each latent modality (Image, Point Cloud, and Proprioceptive State) to the final action generation, we conduct a modality-ablation clustering analysis. As shown in Table 10, removing any single modality from the reasoning space degrades the clustering quality (i.e., reduces the inter-class distance). This confirms that the 2D visual, 3D geometric, and proprioceptive modalities provide indispensable, orthogonal information that is critical for effective and distinguishable action generation.

*Table 10.* **Ablation on Latent Modalities.** The effect of removing individual modalities on the action inter-class distance.

| Latent Modality Configuration | Action Inter-class Distance |
|---|---|
| w/o 2D (3D + State only) | 1.22 |
| w/o 3D (2D + State only) | 1.23 |
| w/o State (2D + 3D only) | 1.30 |
| **All Modalities (LaST$_0$)** | **1.34** |

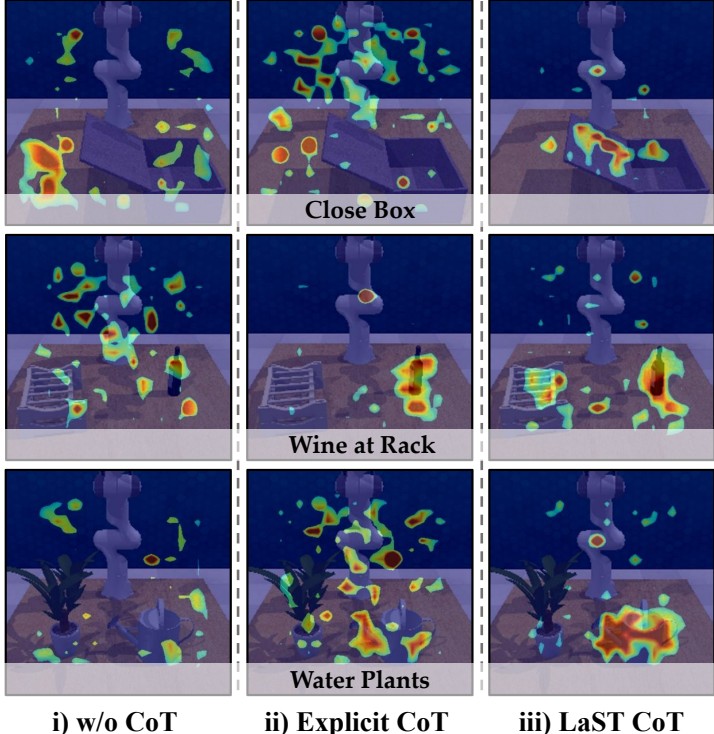

**i) w/o CoT**    **ii) Explicit CoT**    **iii) LaST CoT**

*Figure 9.* **Visualization of attention heatmaps.** We visualize the attention heatmaps from the final layer of LaST$_0$ on RLBench observations. The red area indicates the regions with high attention weights, highlighting the model's focus on task-relevant entities.

## G. Additional Visualization

### G.1. Additional Attention Visualizations.

As shown in Fig. 9, we compare the attention heatmaps of LaST$_0$ against variants without CoT and with explicit CoT (CoT-VLA) on several RLBench tasks. While the No-CoT and Explicit CoT variants fail to attend to the critical dynamics of the scene, often focusing on irrelevant background textures, LaST$_0$ demonstrates precise semantic alignment. It explicitly targets the interaction between the robot arm and the object, highlighting its superior spatio-temporal understanding capabilities.

### G.2. Additional Visualizations on VGGT generated point clouds.

Due to the lack of point cloud data and accurate camera extrinsics in most pretraining datasets, we use VGGT-generated point clouds during the pretraining stage. As shown in Fig. 10, we show the generated point clouds and its corresponded RGB images to provide a clearer view of the noise characteristics and reconstruction quality. The point clouds generated by VGGT provides crucial spatial geometric information for the pretraining stage of LaST$_0$, constructing a complete and generalizable LaST CoT. Meanwhile, to assess the impact of synthetic 3D noise, we replace the ground-truth point clouds in RLBench with VGGT-generated point clouds as 3D latent supervision during downstream task training. The average success rate across 10 tasks only drops marginally from 82% to 81%, indicating that VGGT can provide reasonably accurate 3D latent supervision, while our latent CoT mechanism is also robust to such noise.

### G.3. Additional Real-World Visualizations.

Fig. 12 illustrates representative task executions across single-arm and dual-arm settings, mobile manipulation tasks and dexterous hand tasks. We observe that LaST$_0$ produces smooth and continuous motions, particularly in precise actions such as surface wiping, spatula-mediated placement, and bimanual scooping. This behavior is enabled by the proposed LaST CoT and dual-system design, where a low-frequency reasoning expert provides temporally coherent latent guidance, while the action expert operates at a higher control frequency, allowing rapid and fine-grained response to environment dynamics. We also verify the model's closed-loop capability: for example, when new marks are continuously drawn on the whiteboard during manipulation, the robotic arm can persistently perform the erasing action.

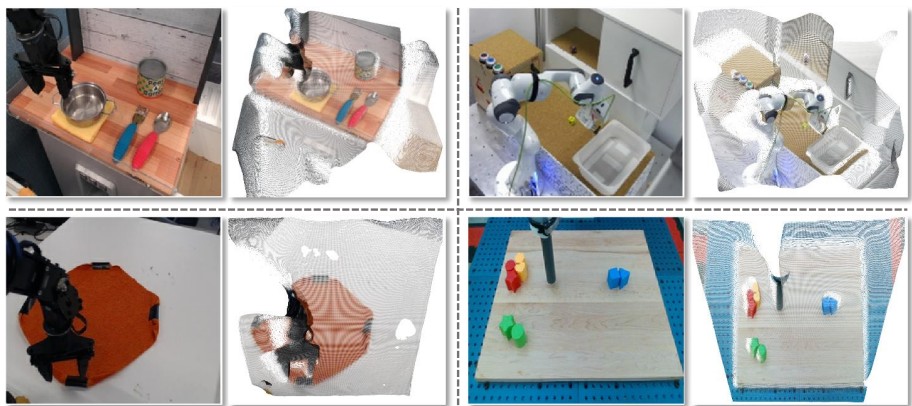

*Figure 10.* **Visualization of VGGT generated point clouds.** We perform a visual comparison between the point clouds generated by VGGT and their corresponding original RGB images; VGGT produced high-quality point clouds embedded with spatial geometric information.

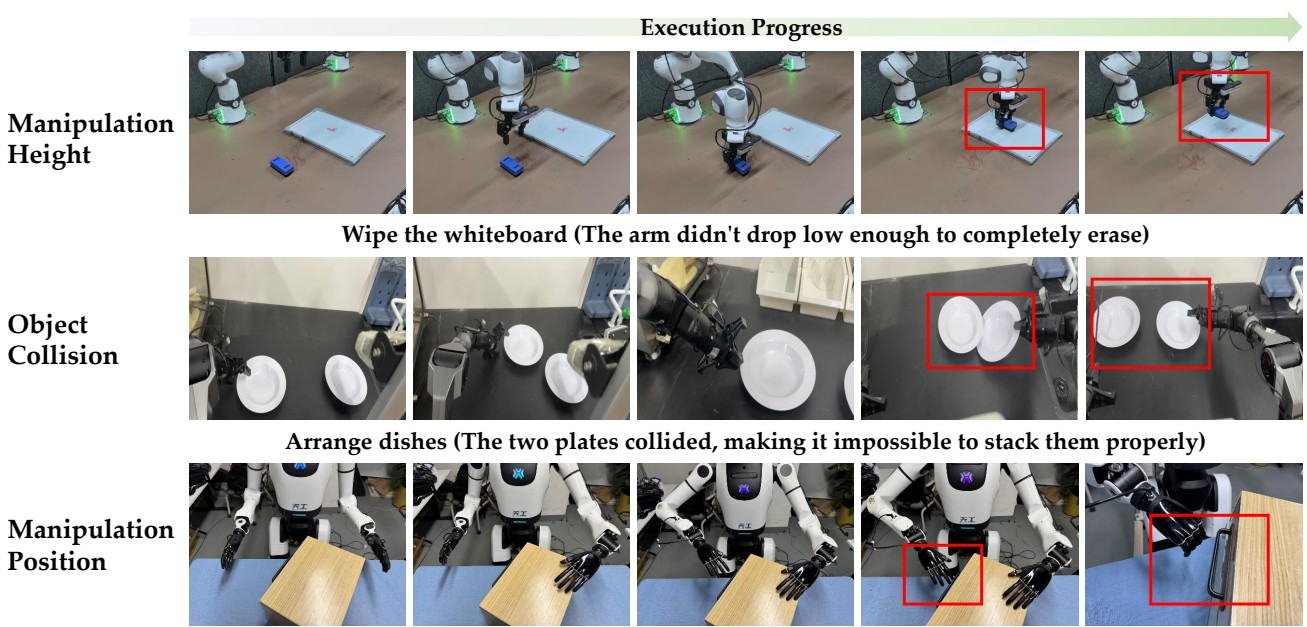

*Figure 11.* Visualization of failure cases on different robot platforms, the task progresses from left to right, and red box highlights the failure positions.

## H. Failure Case Analysis

For different scenarios, we recorded three typical failure cases, as shown in Fig. 11, where the red boxes indicate the specific locations and results of the failures.

1) In the first case, the FR3 arm is required to use an eraser to wipe a pattern off a whiteboard, but due to an error in the **manipulation height**, it did not descend to a height that was completely flush with the whiteboard, resulting in the pattern not being completely erased. The lack of visual feedback during the erasing process regarding the presence or absence of the pattern increases the probability of this happening.

2) The second case demonstrates **object collision** during a dual-arm collaborative task. During the process of the right arm placing the second plate onto the first one, the two plates collided, causing the first plate to shift from its original position. This occurred because the model incorrectly estimated the required height and position for stacking the plates.

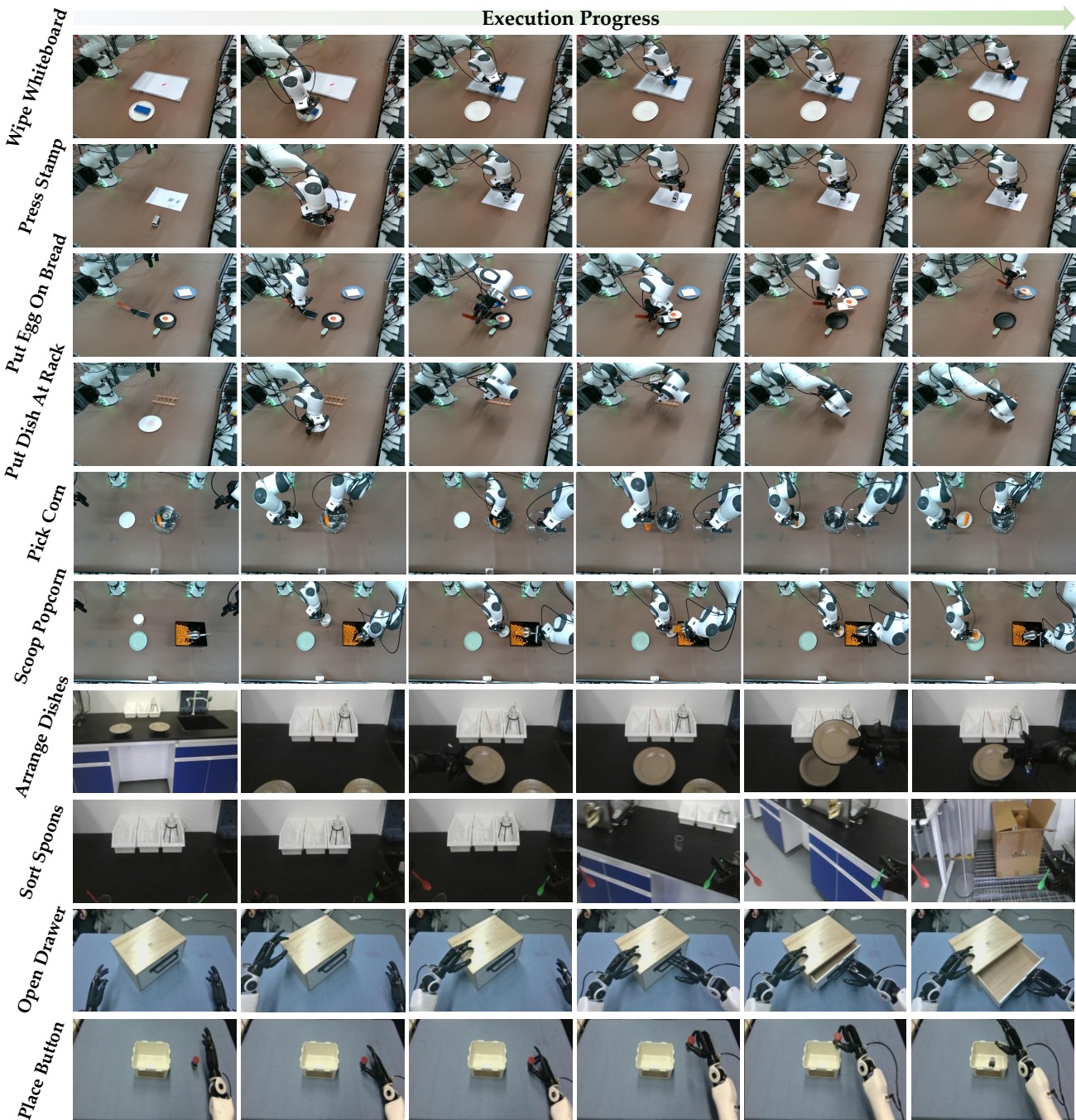

*Figure 12.* Visualization of complete task execution processes by real-world tasks (from left to right).

3) The failure in the third case in the dexterous manipulation task is a typical case of **incorrect manipulation position**. The model did not move the hand forward enough to grasp the drawer handle, leading to a failed pull.

## I. Additional Method Details

### I.1. Additional Preliminaries

**Robotic CoT Reasoning.** To bridge the gap between high-level visual observations and low-level control, recent VLA methods introduce an intermediate CoT variable $\mathcal{Z}$. This formulation decomposes the policy distribution into a reasoning

stage (predicting $\mathcal{Z}$) and an execution stage (predicting $\mathbf{a}$ conditioned on $\mathcal{Z}$):

$$p(\mathbf{a}, \mathcal{Z} \mid I_t, l) = p(\mathbf{a} \mid \mathcal{Z}, I_t, l) \cdot p(\mathcal{Z} \mid I_t, l)$$

Existing methods typically adopt an explicit CoT paradigm, where $\mathcal{Z}$ consists of discrete natural language tokens (Li et al., 2025a) or image tokens (Zhao et al., 2025; Zhang et al., 2025a). While interpretable, these approaches confine reasoning to the linguistic space of LLMs, which struggles to represent ineffable physical attributes and incurs inference latency due to explicit decoding. In this paper, we propose a new latent CoT paradigm for the robotic domain. Unlike explicit CoT, we define $\mathcal{Z} = \{\mathbf{z}_1, \ldots, \mathbf{z}_k\}$ as a sequence of continuous embeddings in a high-dimensional latent space. In our framework, the latent variable $\mathcal{Z}$ is trained to autoregressively predict future dynamics, including latent representations of 2D images, 3D point clouds, and robot proprioceptive states, thereby modeling the physical world in a compact space.

### I.2. Dual-System Coordination

**Inference via KV Cache.** During inference, the Key-Value states computed by the slow expert (encapsulating the Latent CoT) are cached in memory. During intermediate fast-control steps, the acting expert only encodes the current observation and attends to the frozen latent CoT cache, effectively retrieving the CoT tokens with $O(1)$, without re-invoking the slow reasoning process. This approach eliminates repetitive decoding, allowing our model to achieve an inference speed of 15.4 Hz on a single RTX 4090 GPU when operating with a 1:4 fast-slow frequency ratio. In particular, the slow reasoning expert runs at 12.7 Hz, while the fast acting expert runs at 22.1 Hz, resulting in 15.4 Hz in total each 4 steps.

