# OpenReview forum: "LaST$_{0}$: Latent Spatio-Temporal Chain-of-Thought for Robotic Vision-Language-Action Model"
_ICML.cc/2026/Conference — ICML 2026 spotlight_

### Official Review · Reviewer_GaKR · 2026-03-06

**Soundness:** 3
**Presentation:** 3
**Significance:** 3
**Originality:** 2
**Overall Recommendation:** 5
**Confidence:** 4

**Summary:**

This paper introduces $LaST\_0$, a Vision-Language-Action (VLA) model that addresses the high latency and representation bottlenecks of explicit Chain-of-Thought (CoT) reasoning by introducing a Latent Spatio-Temporal CoT. The method leverages a hierarchical, dual-system Mixture-of-Transformers (MoT) architecture split into two experts: a low-frequency reasoning expert that autoregressively predicts continuous latent tokens for future 2D visual, 3D geometric, and robot proprioceptive states, and a high-frequency acting expert that uses these cached representations alongside real-time observations to generate continuous actions via flow matching. Empirically, $LaST\_0$ outperforms baselines on most tasks evaluated from RLBench, which has a significantly better ratio between success rate and inference speed. Real-world deployments across tabletop, mobile, and humanoid dexterous platforms further validate the method’s practicality and robustness.

**Compliance With Llm Reviewing Policy:**

Affirmed.

**Final Justification:**

The paper is in good state pre-rebuttal, and the rebuttal is satisfactory to me. I will maintain my original rating of accept.

**Key Questions For Authors:**

- Include a limitation section.
- The pre-training phase utilizes VGGT to generate synthetic 3D point clouds. How does the quality or inherent noise of this method impact the model's downstream performance? It would be helpful if the authors include visual examples of the generated point clouds in the appendix too.

**Limitations:**

Limitations not discussed.

**Strengths And Weaknesses:**

**Strengths:**
- The Mixture-of-Transformers architecture is an elegant design choice.
- Demonstrates consistent improvement over baselines while maintaining a good balance between performance and inference latency, with thorough evaluation across simulation and real-world settings with multiple embodiments.
- The analysis in Fig. 4 is very useful, providing interpretability of why the method works better.
- Comprehensive ablation studies on modalities, number of latent tokens, temporal coverage, and collaboration frequency in Fig. 5.

**Weaknesses:**
- Limitation not discussed.
- [Presentation] The readability of Figure 5 could be improved. Adding vertical separator lines between the subplots or utilizing distinct color palettes for each individual ablation would help visually distinguish the sub figures.

---

> ### Author Rebuttal · Authors · 2026-03-31
>
> **Dear Reviewer GaKR,**
>
> We sincerely thank you for your time and constructive feedback. We appreciate the recognition of our method as “an elegant design choice,” as well as the acknowledgment that our visualizations “provide interpretability of why the method works better,” and the positive assessment of our experiments as “demonstrating consistent improvements” and “comprehensive ablation studies.” Below, we provide detailed responses to each of your concerns.
>
> ---
>
> ### **[W1 & Q1]. Limitation not discussed**
>
> We will include a dedicated Limitations and Future Work section in the revised version. We summarize the limitations of our work in three aspects.
>
> First, regarding computational cost, VLA models with reasoning capabilities inherently introduce additional overhead. To mitigate this issue, we introduce an asynchronous fast-slow design that fully leverages the representations of our proposed temporal latent CoT, and further employ mixed-precision training to improve efficiency. Nevertheless, optimizing computational efficiency remains an important direction for future VLA model design.
>
> Second, regarding limited pretraining data diversity, our current pretraining data has relatively limited coverage of dexterous and mobile manipulation scenarios. This is largely due to the scarcity of publicly available datasets in these domains. In future work, we plan to incorporate more diverse data sources, including large-scale human egocentric data, to improve pretraining diversity.
>
> Third, as discussed in Appendix E (Line 929), our model shows limitations in handling complex object-to-object interactions, particularly collision reasoning. Although our method reasons about 3D structure in the LaST latent space, we aim to further enhance the latent CoT by explicitly modeling 3D relationships between objects in future work.
>
> ---
>
> ### **[W2]. The readability of Figure 5**
>
> Thank you for the suggestion. Due to rebuttal policy constraints, we are unable to include updated figures here. We will revise Figure 5 in the final version by adding vertical separators for each subfigure to improve clarity.
>
> ---
>
> ### **[Q2]. Synthetic 3D point clouds**
>
> Following your constructive suggestion, we will include additional visualizations of VGGT-generated point clouds in the appendix to provide a clearer view of the noise characteristics and reconstruction quality. Meanwhile, to assess the impact of synthetic 3D noise, we replace the ground-truth point clouds in RLBench with VGGT-generated point clouds as 3D latent supervision during downstream task training. **The average success rate across 12 tasks only drops marginally from 82% to 81%, indicating that VGGT can provide reasonably accurate 3D latent supervision, while our latent CoT mechanism is also robust to such noise.** Due to the lack of point cloud data and accurate camera extrinsics in most pretraining datasets, we use VGGT-generated point clouds only during the pretraining stage.
>
> |Model|Success rate|Source of 3D latent|
> |-|-|-|
> |LaST$_0$|82%|ground truth|
> |LaST$_0$|81%|VGGT|

---

> > ### Author Rebuttal · Reviewer_GaKR · 2026-04-01
> >
> > The authors have addressed all my concerns. I maintain my original rating.

---

> > > ### Author Response · Authors · 2026-04-03
> > >
> > > **Dear Reviewer GaKR,**
> > >
> > > We are pleased that our rebuttal has addressed your concerns. Thank you for your thoughtful feedback and for maintaining your original "**Accept**" rating after the rebuttal. We sincerely appreciate your constructive comments, which have helped us further improve the paper.
> > >
> > > **Best Regards,**
> > >
> > > **The Authors**

---

### Official Review · Reviewer_W7aj · 2026-03-10

**Soundness:** 3
**Presentation:** 3
**Significance:** 4
**Originality:** 4
**Overall Recommendation:** 6
**Confidence:** 4

**Summary:**

This paper proposes a VLA framework that enables efficient reason-before-act behavior through a Latent Spatio-Temporal CoT representation LaST0. Instead of generating explicit textual reasoning or predicting future observations, the method performs reasoning in a compact latent space that captures multimodal physical dynamics. The dual-system design reduces inference latency while maintaining temporally consistent reasoning for robotic manipulation. Experiments on multiple embodiments and tasks, including single-arm, dual-arm manipulation, mobile manipulation, and dexterous hand manipulation, demonstrate the effectiveness and superior performance of LaST0.

**Compliance With Llm Reviewing Policy:**

Affirmed.

**Final Justification:**

The authors have successfully addressed my concerns. I will raise my rating from Accept to Strong Accept.

**Key Questions For Authors:**

Please look at the above mentioned weaknesses.

**Limitations:**

yes

**Strengths And Weaknesses:**

**Strength**

1. The paper introduces a spatio temporal latent chain of thought framework for vision language action models, which enables reasoning in a compact latent space rather than through explicit textual reasoning. This design improves reasoning efficiency and establishes a promising new paradigm for unifying robotic reasoning with action generation.
2. The use of latent chain of thought to capture interactions between a robot and the physical environment is intuitive and well justified. The mixture of transformers based dual system architecture effectively conditions action prediction on latent reasoning while preserving the capacity for real time control.
3. The method is evaluated across multiple embodiments and task settings, including single arm, dual arm, mobile, and dexterous manipulation. The results show clear and consistent quantitative improvements. Performance on long horizon tasks further highlights the value of temporal reasoning.

**Weakness**

1. The paper follows the frame sampling protocol in [1] and mentions keyframe extraction, but it remains unclear whether a comparison between dense frames and keyframes was conducted for latent representation construction. An ablation study on this design choice would provide stronger justification for the adopted setting.
2. Most simulation experiments are conducted only on RLBench under the keyframe setting. The paper would be strengthened by including results from additional simulators under a dense frame setting, which would provide broader evidence of generalization and further validate the advantage of LaST0 on long horizon tasks in simulation.
3. The latent sequence interleaves multiple modalities, including visual, geometric, and robot state tokens, but the effect of modality ordering is not analyzed. Since the paper argues that latent chain of thought captures the relationship between the robot and the environment, it would be valuable to examine whether different modality orderings influence performance or the quality of the learned representations.

[1] Mohit Shridhar, Lucas Manuelli, Dieter Fox. Perceiver-Actor: A Multi-Task Transformer for Robotic Manipulation. In Proc. of CoRL. 2022, 785-799.

---

> ### Author Rebuttal · Authors · 2026-03-31
>
> **Dear Reviewer W7aj,**
>
> We appreciate the reviewer’s recognition that our method “establishes a promising new paradigm for unifying robotic reasoning with action generation” and is “intuitive and well justified.” We are also grateful for the reviewer’s positive assessment of our experiments as “showing clear and consistent quantitative improvements.” Below, we provide detailed responses to each of the reviewer’s concerns.
>
> ---
> ### **[W1] Ablation Study of Latent Keyframe Design**
> Following your constructive comments, we conduct an additional ablation study to investigate the impact of using keyframe-based latent representations versus densely sampled latent representations in LaST CoT for manipulation. To this end, we perform experiments on the LIBERO benchmark, a widely used simulation platform that operates on high-frequency, dense observations across diverse tasks. For these experiments, we introduce a new hyperparameter **latent stride**, which enforces temporal sparsity in the latent reasoning process while maintaining dense action outputs. We validate the effectiveness of this design through an ablation study on 10 tasks from the LIBERO-Spatial benchmark, analyzing the impact of latent temporal spacing on performance. Note that we use the official training data provided by the benchmark, and the training implementation details are consistent with those described in the submission, except that we additionally incorporate wrist-view observations in the action expert. As shown in the table below, maintaining a sparse latent construction (latent stride = 8) is critical in continuous prediction settings, as it enables the reasoning expert to capture more informative long-horizon dynamics. These results show that sparse latent construction (keyframe-like setting) is not only sufficient but also beneficial, as it enables the model to focus on long-horizon dynamics rather than redundant frame-level details. Due to time constraints, we will extend this ablation study to more tasks in the revised version.
> latent_stride (frames) |1|2|4|8|16
> -|-|-|-|-|-
> Success Rate(%)|97.4|98.0|98.8|99.2|99.0
>
> ---
> ### **[W2] Experiments on Other Simulators**
> **a)** First, in RLBench simulation, our use of a keyframe-based setting follows prior works on this benchmark (Shridhar et al., 2022; Goyal et al., 2023). However, this does not imply that our method is limited to keyframe-based settings or is not applicable to dense-frame scenarios. To further validate this, we conduct additional experiments on the **LIBERO benchmark**. For fair comparison, all methods use identical input views. As detailed in the table below, LaST₀ achieves SOTA performance across all four LIBERO task suites, obtaining an impressive average success rate of **98.1%**.
> Model|spatial|object|goal|long|average
> -|-|-|-|-|-
> OpenVLA|84.7|88.4|79.2|53.7|76.5
> CoT-VLA|87.5|91.6|87.6|69.0|81.1
> DreamVLA|97.5|94.0|89.5|89.5|92.6
> $\pi_{0}$|96.8|98.8|95.8|85.2|94.1
> $\pi_{0.5}$|98.8|98.2|98.0|92.4|96.9
> OpenVLA-OFT|97.6|98.4|97.9|94.5|97.1
> LaST$_0$|99.2|99.6|98.0|95.6|98.1
>
> **b) Regarding long-horizon capability**, as reported in Table 2 and Lines 422–435, LaST₀ consistently outperforms $\pi_{0.5}$ across three long-horizon real-world tasks, including sequential tabletop manipulation and mobile dual-arm manipulation, with success rate improvements of +26%, +10%, and +7%, respectively. This widening performance gap demonstrates LaST$_0$'s ability to learn temporally coherent latent representations and reason effectively over extended horizons.
> Furthermore, as shown in the table above, our method also achieves outstanding performance on the LIBERO-Long benchmark, which consists of a series of long-horizon tasks. This further validates the robustness of our approach across both real-world and simulated environments.
>
> ---
> ### **[W3] Modality Ordering**
> Following your insightful suggestion，we further conduct an ablation study on modality ordering on the 12-task RLBench benchmark. In our original design, the latent sequence follows an environment-to-robot cognitive flow: **[Visual, Geometric, State]**. To rigorously evaluate whether this ordering affects the quality of latent representations and manipulation performance, we examine multiple permutations of these modalities.
> Modality Ordering (Latent Sequence)|Success Rate (%)
> -|-
> [Visual, Geometric, State] (LaST0)|82
> [State, Geometric, Visual]|78
> [Geometric, Visual, State]|79
> [Visual, State, Geometric]|81
> [State, Visual, Geometric]|80
> [Geometric, State, Visual]|79
>
> The results show that the model remains robust across different permutations, while the original ordering consistently achieves the best manipulation performance. This validates our design choice of the **[Visual, Geometric, State]** sequence, which aligns with a natural embodied reasoning process: first establishing global semantic context, then refining it with spatial structure, and finally grounding it in the robot’s physical state before action prediction.

---

> > ### Author Rebuttal · Reviewer_W7aj · 2026-04-03
> >
> > The authors have successfully addressed all the concerns raised in my review. I am satisfied with their responses and will raise my rating.

---

> > > ### Author Response · Authors · 2026-04-03
> > >
> > > **Dear Reviewer W7aj,**
> > >
> > > We are glad that our rebuttal has successfully addressed your concerns. We truly appreciate your time and thoughtful evaluation, as well as your decision to raise your rating to "**strong accept**". This is a strong recognition of our work and efforts.
> > >
> > > **Best Regards,**
> > >
> > > **The Authors**

---

### Official Review · Reviewer_nzZo · 2026-03-10

**Soundness:** 3
**Presentation:** 3
**Significance:** 3
**Originality:** 2
**Overall Recommendation:** 5
**Confidence:** 4

**Summary:**

This paper introduce a VLA model that uses latent CoT reasoning to predict future 2D, 3D and proprioceptive states. LaST uses a MoT architecture with a low-frequency reasoning model and a high-frequency action model, achieving fast inference and good performance.

**Compliance With Llm Reviewing Policy:**

Affirmed.

**Final Justification:**

The authors rebuttal, particularly the PCA analysis, is convincing.

**Key Questions For Authors:**

* Most training data in the split is single-arm data (bridge 20%, fractal 13%, kuka 20%) and the dual-arm or humanoid data is sparse (RoboMIND 0.2%). Under this setting, how can this model perform well in dual-arm and humanoid setting?
* The method uses only one token for future 2D/3D/proprio representation and the single token is from pooling. How could this be effective considering the tiny difference between consecutive frames? Visualization about the latent codes is needed (PCA, attention maps from the visual encoder, etc.).

**Limitations:**

See Questions.

**Strengths And Weaknesses:**

* The motivation of the paper is significant. CoT is very important for some embodied tasks. Current methods face many challenges such as real-time inference and the alignment between CoT and actions.
* The dual system is very innovating. Use low-frequency model for reasoning and high frequency model for action generation.This keeps the strong ability of LLMs and hide the reasoning time for real-time excution. Also, using the latent code as an information bridge is a good design.

---

> ### Author Rebuttal · Authors · 2026-03-31
>
> **Dear Reviewer nzZo,**
>
> We appreciate the recognition of our work, including that “the motivation of the paper is significant,” “the dual system is very innovative,” and “the latent code as an information bridge is a good design.” Below, we provide detailed responses to each of concerns.
>
> ---
> ### **[Q1] Pre-training data**
>
> Thank you for the insightful question. We address this concern from three aspects:
>
> **a) Efficient transfer via minimal reinitialization.** During large-scale pretraining, the VLA learns embodiment-agnostic representations of the physical world. When adapting to new downstream tasks with different action DoFs, we only reinitialize a small fraction of parameters (i.e., the action projection MLPs), while fully reusing the pretrained backbone weights. This enables rapid and effective adaptation to downstream data. This is consistent with prior works that leverage human data for pretraining: although the control DoFs between humans and robots may not align, such pretraining still enables VLA models to learn representations that generalize to robotic scenarios.
>
> **b) Advantage of our proposed Latent Spatio-Temporal (LaST) CoT.** LaST CoT operates in a structured spatio-temporal latent space that jointly models semantic, geometric, and proprioceptive dynamics. As shown in Fig. 4, our model better captures interactions between the robot and its environment, decouples physical reasoning from embodiment-specific action spaces, and enables effective transfer under new data distributions. To further validate this, we replace our backbone with another SOTA VLM (Qwen3-VL-4B, without additional robotic pretraining) and evaluate it on 12 RLBench tasks under identical settings. Our method maintains SOTA-level performance with the Qwen3 backbone (76% success rate), indicating that our LaST CoT design improves the model’s physical modeling and action learning capabilities regardless of the pretraining data composition, even without robotic data pretraining.
>
> **c) Fairness of pretraining data.** High-quality pretraining data for mobile and dexterous manipulation remains scarce. Prior VLA methods (e.g., OpenVLA, CogACT, HybridVLA) also include limited amounts of such data during pretraining. To ensure a fair comparison, we carefully align our pretraining data distribution with theirs. This design isolates the contribution of our LaST CoT and model design from potential advantages arising from differences in pretraining data scale.
>
> ---
> ### **[Q2] Latent reasoning space**
>
> **a)** Following the reviewer’s suggestion, we conduct **Principal Component Analysis (PCA) on the latent CoT tokens** in the RLBench 12 tasks data. As shown in the table below, PCA yields well-separated clusters across all modalities, reflecting clear inter-token discrimination in the latent space. We further relate this analysis to our token-number ablation, which varies the number of tokens per modality at each time step. We find that increasing the number of tokens used to represent the latent space yields only marginal gains in inter-token distance. This result aligns with the experimental findings in Fig. 5(b) of our submission, suggesting that a single token per modality is sufficient to capture the essential physical dynamics and temporal variation. Due to rebuttal constraints, we report only quantitative inter-class distance results here, while visualizations will be included in the revised version.
>
> **Table A. Consecutive Frames Inter-Distance by PCA**
> ||Image|||Point Cloud|||State|
> |-|-|-|-|-|-|-|-|
> ||1 Token|4 Tokens|16 Tokens|1 Token|4 Tokens|16 Tokens|1 Token|
> |Step (t vs t+1)|0.68|0.70|0.63|0.20|0.20|0.29|0.57|
> |Step (t vs t+2)|0.78|0.79|0.69|0.41|0.33|0.42|0.19|
> |Step (t vs t+3)|0.97|0.90|0.88|0.52|0.42|0.55|0.50|
> |Step (t vs t+4)|1.38|1.30|1.22|1.09|1.15|1.16|0.83|
>
> **b)To assess the effect of the latent CoT,**  we perform a PCA-based quantitative analysis of action token features, by projecting them into 2D space and measuring inter-class distances between motion categories. As shown in the table below, we observe that our model produces larger inter-class separation compared to baselines. This indicates that the latent representations are more discriminative and structured, leading to more stable and consistent action generation.
> |Model|Action Inter-class Distance|
> |-|-|
> |w/o LaST CoT (Baseline)|1.03|
> |LaST$_0$|1.34|
>
> **c) To explicitly analyze each latent modality's contribution to action generation,** we conduct a PCA-based clustering ablation. As shown below, removing any single modality degrades clustering quality, confirming that each provides indispensable, orthogonal gains critical for effective action generation.
>
> |Latent Modality Configuration|Action Inter-class Distance|
> |-|-|
> |w/o 2D (3D + State only)|1.22|
> |w/o 3D (2D + State only)|1.23|
> |w/o State (2D + 3D only)|1.30|
> |All Modalities (LaST$_0$)|1.34|

---

> > ### Author Rebuttal · Reviewer_nzZo · 2026-04-02
> >
> > The authors have addressed all my concerns.

---

> > > ### Author Response · Authors · 2026-04-03
> > >
> > > **Dear Reviewer nzZo,**
> > >
> > > We are pleased that our rebuttal has successfully addressed your concerns. Thank you for your recognition of our work and for increasing your score to "**accept**". This is a strong recognition of our work and efforts.
> > >
> > > **Best Regards,**
> > >
> > > **The Authors**

---

### Official Review · Reviewer_XPMK · 2026-03-15

**Soundness:** 2
**Presentation:** 3
**Significance:** 3
**Originality:** 2
**Overall Recommendation:** 4
**Confidence:** 4

**Summary:**

This paper proposes $LaST_0$, a Vision-Language-Action (VLA) model that introduces a Latent Spatio-Temporal Chain-of-Thought (LaST CoT) to enable efficient reason-before-act behavior in robotic manipulation. The method performs reasoning in a multimodal latent space that models future visual, geometric, and proprioceptive states. A dual-system architecture coordinates low-frequency reasoning with high-frequency action generation. Experiments on RLBench and multiple real-world robot platforms demonstrate improvements in both task success rate and inference efficiency over several existing VLA baselines.

**Compliance With Llm Reviewing Policy:**

Affirmed.

**Final Justification:**

The authors have addressed all my concerns.

**Key Questions For Authors:**

None

**Strengths And Weaknesses:**

## Strengths
1. The idea of performing reasoning in latent space aims to reduce the latency associated with explicit CoT reasoning.

2. The dual-system architecture separates reasoning and action generation, providing a clear framework for integrating deliberative reasoning with real-time control.

3. The evaluation spans RLBench and multiple physical robot platforms, demonstrating potential practical applicability.

4. The paper studies the impact of modality combinations, temporal coverage, and collaboration frequency between reasoning and action modules.



## Weaknesses

1. **Limited methodological novelty.**
   The core idea of the paper is to perform reasoning in a latent space and coordinate reasoning with action generation via a dual-system architecture. However, the framework mainly combines several existing techniques, including latent reasoning, Transformer-based VLA architectures, multimodal representations, and flow-based action generation. The paper does not clearly demonstrate that this combination leads to fundamentally new algorithmic insights rather than an engineering integration of existing components.

2. **Insufficient ablation studies.**
   The ablation experiments are mainly conducted in simulation, and only a limited set of design variations are explored. More comprehensive combinations of ablations (e.g., latent modalities, temporal horizon, reasoning frequency, architecture components) are not systematically evaluated. As a result, the individual contribution of each component to the final performance improvement remains unclear.

3. **Limited evaluation metrics.**
   The experiments mainly rely on task success rate as the primary metric. More comprehensive evaluations of model robustness, stability, few-shot generalization, or long-horizon execution consistency are missing. In addition, the paper does not evaluate the method on more complex long-horizon manipulation tasks where reasoning capability could be more clearly demonstrated.

4. **Missing comparisons with closely related approaches.**
   The experiments mainly compare with several VLA baselines. However, there is limited comparison with recent world-model-based that also attempt to model future dynamics in latent space. Without such comparisons, it is difficult to clearly assess the relative advantages of the proposed method.

5. **Limited interpretability of the latent reasoning space.**
   The paper proposes reasoning in a multimodal latent space, but it provides limited analysis or visualization showing how these latent representations influence action generation. It remains unclear what information each modality contributes or how the latent reasoning evolves during task execution.

6. **Limited reproducibility.**
   Important training details are not fully specified, and the paper does not provide code or implementation details. This raises concerns about the reproducibility of the reported results.

7. **Limited exploration of backbone model choices.**
   The proposed architecture is only evaluated using DeepSeek-LLM 1.5B as the backbone. It is unclear whether the improvements are specific to this backbone. For example, it would be valuable to evaluate whether similar improvements could be achieved when integrating the proposed architecture into other strong VLA backbones such as π0.5.

8. **Large-scale pretraining data may confound performance gains.**
   The model is pretrained on a large corpus of robotic datasets. It is therefore difficult to disentangle whether the performance gains come from the proposed architectural design or from the scale and diversity of the training data.

9. **Questions regarding embodiment-specific action encoding.**
   The paper evaluates different robotic embodiments with varying action spaces. However, it is unclear whether the action encoder is reinitialized for each embodiment and whether training steps or optimization settings are kept consistent. These factors may affect the fairness and interpretation of the results.

---

> ### Author Rebuttal · Authors · 2026-03-31
>
> **Dear Reviewer XPMK,**
>
> ### **[W1] Novelty**
> Thank you for the detailed comment. We clarify that the key novelty of LaST₀ is a principled solution to a core challenge: achieving physically grounded reasoning with real-time control.
> - Unlike prior CoT approaches with high latency and static paradigms, LaST CoT is the first to capture semantic and geometric dynamics over time within a compact latent space. This design achieves SOTA performance across diverse embodiment while improving the model’s understanding of robot–environment relationship (Fig. 4 of submission).
> - Unlike prior fast–slow designs that simply assign heterogeneous frequencies, our MoT-based dual system leverages temporally extended latent representations from LaST CoT and enables efficient collaboration via shared attention, achieving both long-horizon reasoning and real-time control.
> - Transformer-based VLA architectures and flow-based action generation are standard VLA components (e.g., OpenVLA, $\pi_0$); our contribution lies in a novel robotics-specific LaST CoT integrated with an temperal latent driven dual system for efficient, physically grounded “reason-before-act” behavior.
> ### **[W2] Ablation studies**
> Following your constructive suggestion, we expand design variations on latent modalities (A), temporal horizon (B), reasoning frequency (C), and architecture components (D) on the 12-task RLBench benchmark, along with real-world studies (E). Overall, these results further confirm that the performance gains stem from the proposed design. We will include all results in the revised version.
>
> A. Latent modalities. In Lines 347–362, we evaluate single modalities and combinations. We further include the missing 3D+State (77% success rate, SR). We also conduct an ablation on modality ordering; due to space constraints, please refer to `Reviewer W7aj (W3)`.
>
> B. Temporal horizon. We previously evaluated temporal horizons of 0, 1, 2, 4, 5, and 6 steps (Figure 5c, Lines 377–384). We now include the missing 3-, 7-, and 8-step settings, which achieve SR of 81%, 80%, and 78%, respectively.
>
> C. Reasoning frequency. We originally reported frequency ratios of 1:1, 1:2, 1:4, 1:8, and the mixed strategy (Figure 5d). We have further included 1:3, 1:5, 1:6, and 1:7 ratio, which achieve SR of 76%, 78%, 76%, and 76%, respectively.
>
> D. Architecture Components. We conduct an additional ablation to evaluate our MoT dual-system design. Compared to a single-expert baseline, where one LLM jointly generates latent representations and actions (74% SR), our method achieves an 8% higher SR.
>
> E. Real world ablations. We conduct ablation studies on the role of different modalities in the “place egg on bread” task. The results are consistent with the trends observed in simulation.
> Modality Combination|SR (%)
> -|-
> 2D+3D|53
> 2D+State|60
> 3D+State|60
> 2D+3D+State|66
> ### **[W3] Evaluation metrics**
> We incorporate additional metrics to evaluate model robustness, stability, and long-horizon consistency.
>
> (a) We evaluate real-world generalization on “place egg on bread” under unseen objects (egg→lemon) and background distractors (15 rollouts each). LaST₀ outperforms π₀.₅, showing strong robustness to unseen scenarios. To evaluate stability, we conduct three test runs and report the resulting variance.
> ||π₀.₅|LaST₀
> -|-|-
> Original|0.47|0.66
> Unseen Objects|0.27±0.05(-43%)|0.45±0.03(-32%)
> Complex Backgrounds|0.33±0.05(-29%)|0.57±0.04 (-14%)
>
> (b) In Table 2 and Lines 422–435, our method demonstrates consistent long-horizon performance on three real-world evaluations, including the “place egg on bread” and mobile manipulation tasks. Please refer to `Reviewer W7aj (W2, b)` for additional long-horizon simulation results.
>
> ### **[W4] More baselines**
> Beyond CoT-VLA (66%) in our submission, we also compare with two recent world-model-based policies, DreamVLA (65%) and Dreamzero (72%), under the same RLBench setting. Our method (82%) consistently outperforms these approaches.
> ### **[W5] Interpretability of the latent space**
> We complement Fig. 4 (attention visualization) with a PCA to examine the impact of latent CoT. Please refer to `Reviewer nzZo (Q2)`, removing LaST CoT or any latent modality degrades clustering quality, indicating each provides indispensable signals for action generation.
> ### **[W6]**
> In Line 319-329+, we provide key implementation details. In revised version, we will release the code and model checkpoints on GitHub.
>
> ### **[W7]**
> We replace DeepSeek-LLM 1.5B with Qwen3-VL-4B and evaluate on RLBench under identical settings. Results (76%) show maintained SOTA performance, demonstrating strong cross-backbone generalization.
>
> ### **[W8]**
> Using our identical pretraining data, LaST₀ (82%) still outperforms π₀.₅ (62%, trained from VLM pretraining) on RLBench, showing that the gains stem from our design rather than the data.
>
> ### **[W9]**
> We reinitialize only the action encoder/decoder (MLP) for each embodiment, while all other settings are kept consistent.

---

> > ### Author Rebuttal · Reviewer_XPMK · 2026-04-03
> >
> > The authors have addressed all my concerns.

---

> > > ### Author Response · Authors · 2026-04-07
> > >
> > > Dear Reviewer XPMK,
> > >
> > > We are pleased that our rebuttal has successfully addressed your concerns. Thank you for your recognition of our work and for increasing your score to "**weak accept**". We will further refine the paper in the revised version.
> > >
> > > Best Regards,
> > >
> > > The Authors

---

### Decision · Program_Chairs · 2026-04-30

**Decision:**

Accept (spotlight)

**Comment:**

This paper introduces a latent spatio-temporal Chain-of-Thought for VLA models, replacing text-based reasoning with predicted multimodal latent tokens that capture future visual features, 3D structure, and robot state. It uses a dual-system Mixture-of-Transformers that combines a slow reasoning module with a fast action module through shared attention, enabling “reason-before-act” without the overhead of language-based CoT.

Reviewers broadly found this work positive. They highlighted the dual-system setup as novel and the latent representation as a clean way to connect reasoning and action. The overall design was seen as intuitive and well-motivated. Experimentally, LaST_0 shows solid gains on RLBench, strong LIBERO performance, and good results on real-world tasks, outperforming prior VLA and world-model baselines. The rebuttal made the paper stronger by adding LIBERO results, showing transfer to another backbone, more ablations, robustness to noisy point clouds, and better interpretability analysis.

I recommend acceptance. In the final version, the authors should include the rebuttal results into the main paper, add a limitations section, and release code as promised.